# Hydrophobic mismatch drives self-organization of designer proteins into synthetic membranes

Justin A. Peruzzi [1,2], Jan Steinkühler [2,3], Timothy Q. Vu[2,3], Taylor F. Gunnels [2,3], Vivian T. Hu [2,3], Peilong Lu [4,5,6], David Baker [7,8,9] & Neha P. Kamat [2,3,10] ✉

The organization of membrane proteins between and within membrane-bound compartments is critical to cellular function. Yet we lack approaches to regulate this organization in a range of membrane-based materials, such as engineered cells, exosomes, and liposomes. Uncovering and leveraging biophysical drivers of membrane protein organization to design membrane systems could greatly enhance the functionality of these materials. Towards this goal, we use de novo protein design, molecular dynamic simulations, and cell-free systems to explore how membrane-protein hydrophobic mismatch could be used to tune protein cotranslational integration and organization in synthetic lipid membranes. We find that membranes must deform to accommodate membrane-protein hydrophobic mismatch, which reduces the expression and co-translational insertion of membrane proteins into synthetic membranes. We use this principle to sort proteins both between and within membranes, thereby achieving one-pot assembly of vesicles with distinct functions and controlled split-protein assembly, respectively. Our results shed light on protein organization in biological membranes and provide a framework to design self-organizing membrane-based materials with applications such as artificial cells, biosensors, and therapeutic nanoparticles.

Biological cells leverage membrane-bound compartments to perform complex functions with precise spatial and temporal control. To execute these processes, cells must insert and sort proteins into distinct membrane compartments. Cellular membranes possess a variety of mechanisms to control membrane protein location. Protein transport is largely mediated by different protein-protein interactions and protein machinery such as clathrin, COPI, and SNARE proteins[1,2]. However, more passive lipid-protein interactions also play a significant role in driving inter- and intramembrane membrane protein organization[3–11]. Membranes and integral membrane-proteins have been shown to possess complementary physiochemical properties, likely allowing for proper protein sorting and function within cells[5,12,13]. Specifically, protein transmembrane domain length and geometry have been shown to correlate with protein localization between and within different

[1]Department of Chemical and Biological Engineering, Northwestern University, Evanston, IL 60208, USA. [2]Center for Synthetic Biology, Northwestern University, Evanston, IL 60208, USA. [3]Department of Biomedical Engineering, Northwestern University, Evanston, IL 60208, USA. [4]Key Laboratory of Structural Biology of Zhejiang Province, School of Life Sciences, Westlake University, Hangzhou, Zhejiang, China. [5]Westlake Laboratory of Life Sciences and Biomedicine, Hangzhou, Zhejiang, China. [6]Institute of Biology, Westlake Institute for Advanced Study, Hangzhou, Zhejiang, China. [7]Department of Biochemistry, University of Washington, Seattle, WA 98195, USA. [8]Institute for Protein Design, University of Washington, Seattle, WA 98195, USA. [9]Howard Hughes Medical Institute, University of Washington, Seattle, WA 98195, USA. [10]Chemistry of Life Processes Institute, Northwestern University, Evanston, IL 60208, USA. ✉e-mail: nkamat@northwestern.edu

cellular membranes[12–14]. These studies suggest that physical features of membranes and proteins, such as the hydrophobic thickness of transmembrane domains and lipid bilayers, govern the organization of membrane proteins into distinct organelle membranes, thereby controlling membrane-based behaviors.

Despite membrane organization being critical to cellular function, it remains a challenge to replicate this organization in membrane-based materials (e.g., engineered cells, biosensors, and drug delivery vehicles). In many of these materials, lipid membranes play a prominent role, serving as the material scaffold of particles ranging from engineered cells[15,16] and exosomes to lipid nanoparticles[17–19]. However, in many cases, proteins localized to these structures are typically applied in a uniform, surface-adhered manner, limiting the potential of the resulting structures to mimic the spatial complexity and functions (binding, transport, and signaling) of living cells. Engineering membrane protein organization within these lipid structures could enable the design of membrane-based nanoparticles with specialized targeting, transport, and transmembrane signaling capabilities that far exceed their current capabilities. Inspired by the organization of membranes and proteins in natural systems, we hypothesized that we could engineer distinct self-organizing membrane-based materials by designing proteins and membranes with complementary properties. Specifically, we sought to leverage recent evidence that variations in the hydrophobic thickness of membrane proteins and the lipid membranes they reside within, should effectively drive protein positioning[12,13]. This bottom-up approach to assemble protein-integrated membranes should enable the design of a distinct class of membrane-based materials[15,20–28] with properties beyond what is possible in nature.

Towards this goal, we harnessed recent advances in de novo protein design and membrane-augmented cell-free protein synthesis systems to explore how protein and lipid properties affect membrane protein synthesis, integration, and dynamics in a highly controlled environment. Specifically, we designed alpha-helical multipass transmembrane proteins ranging from 10 to 50 Å in hydrophobic length, and transmembrane pores ranging from 20 to 50 Å in hydrophobic length based on previous protein design scaffolds[29,30], and assessed the ability of these proteins to insert and organize into thin (14:1 PC, 23 Å), medium (18:1 PC, 29 Å), and thick lipid membranes (22:1 PC, 37 Å) (Supplementary Table 1, membrane thicknesses measured by Heberle et al. (2020))[31]. Together with molecular dynamic (MD) simulations, and experimental studies using cell-free systems, we were able to systematically probe and characterize how membrane-protein hydrophobic mismatch affects protein expression, co-translational folding, and location within a membrane outside the confines of a living cell. This study provides a route to begin to organize membrane proteins in cell-free systems, an initial step to engineering more complex membrane-based materials.

## Results

### Hydrophobic mismatch reduces co-translational insertion of designed proteins

We first assessed how hydrophobic mismatch, defined here as membrane hydrophobic thickness minus the protein hydrophobic thickness, affects co-translational insertion of de novo designed hairpin proteins (Fig. 1A). We performed coarse-grained MD simulations of proteins in a thin, medium, and thick lipid system (DyPC, DOPC, DGPC, respectively) and measured membrane thickness as a function of distance from the protein we found that membranes deform close to the protein insertion site (Fig. 1B and Supplementary Fig. 1). We then compared the change in membrane thickness (defined as membrane compression) in each lipid system for three membrane proteins, which varied in hydrophobic thickness from 20 to 50 Å, by calculating the difference between the membrane thickness at the protein surface and the membrane thickness sufficiently far away from the protein. We

found that membranes must deform more as hydrophobic mismatch increases, consistent with MD simulations of gramicidin A channels (Fig. 1C)[32].

We next experimentally characterized how hydrophobic mismatch impacted protein expression in vitro. We designed plasmids encoding hairpin proteins. A C-terminal monomeric-enhanced GFP (mEGFP) allowed us to monitor expression and proper folding of proteins by GFP fluorescence[33,34]. By adding a plasmid encoding a membrane protein and pre-assembled phospholipid vesicles to a cell-free protein synthesis system, we could track expression and cotranslational insertion of designed proteins into synthetic membranes of the vesicles (Fig. 1D and Supplementary Fig. 2a, b). The fluorescence of a C-terminal mEGFP has previously been used to assess the insertion and folding of cell-free expressed membrane proteins. However, to validate that this assay could be applied to our computationally designed proteins, we first confirmed that increases in GFP fluorescence correlated to protein insertion. We expressed soluble mEGFP and the 28 Å thick protein in the presence of no membrane condition or a thin, medium, and thick lipid membrane (14:1 PC (23 Å), 18:1 PC (DOPC, 29 Å), and 22:1 PC (37 Å), respectively) and monitored fluorescence changes during expression (Supplementary Fig. 2a, b). We found that soluble mEGFP expression did not significantly change when expressed in the presence of vesicles, however, the expression of the 28 Å thick protein greatly increased when vesicles were added to the cell-free reactions, indicating the membrane protein expression benefitted from interactions with the membranes present in the reaction. We then performed size exclusion chromatography on these samples to separate vesicle-associated and free protein. We only observed co-elution of mEGFP and vesicles when mEGFP was fused to the 28 Å thick protein, confirming that we were able to cotranslationally insert de novo designed proteins into synthetic membranes using a cell-free expression system (Supplementary Fig. 2c–f). Further, by quantifying the GFP fluorescence relative to lipid fluorescence, we can calculate relative amounts of protein per vesicle. These values mirror those measured prior to purification of vesicles, suggesting that increases in GFP fluorescence are reflective of protein expression and integration into synthetic membranes (Supplementary Fig. 2g, h). To further confirm cell-free expressed transmembrane protein insertion into membranes, confocal microscopy was performed (Supplementary Fig. 3).

Once we confirmed that de novo designed proteins could be inserted into synthetic membranes, we expressed an expanded panel of proteins which ranged in hydrophobic thickness from 10 to 50 Å in the presence of no membrane or a thin, medium, or thick lipid membrane (14:1 PC (23 Å), 18:1 PC (DOPC, 29 Å), and 22:1 PC (37 Å), respectively). We monitored protein folding via GFP fluorescence and measured protein expression via western blots following centrifugation. Because the expression of each construct differed, we normalized mEGFP fluorescence or western blot band intensity by the maximum expression for each construct to compare how expression varied in the presence of each lipid system (Fig. 1E and Supplementary Figs. 4 and 5). All designed proteins expressed poorly in the absence of vesicles, suggesting that their enhanced expression in the presence of membranes is due to co-translational protein folding and insertion into synthetic membranes (Fig. 1E and Supplementary Fig. 4). When comparing the expression of each protein in the presence of the three lipid systems, we found that expression and proper folding was generally maximized when membrane-protein hydrophobic mismatch was minimized for each studied protein (Fig. 1E). Interestingly, the smaller 10 Å protein did not follow this trend. We attribute this behavior to its small hydrophobic transmembrane domain, which is less than half the thickness of the thinnest membrane system (23 Å thick), and likely results in the protein behaving more like a soluble protein relative to the other membrane proteins studied. Once we assessed how protein expression varied experimentally, we compared them to simulation

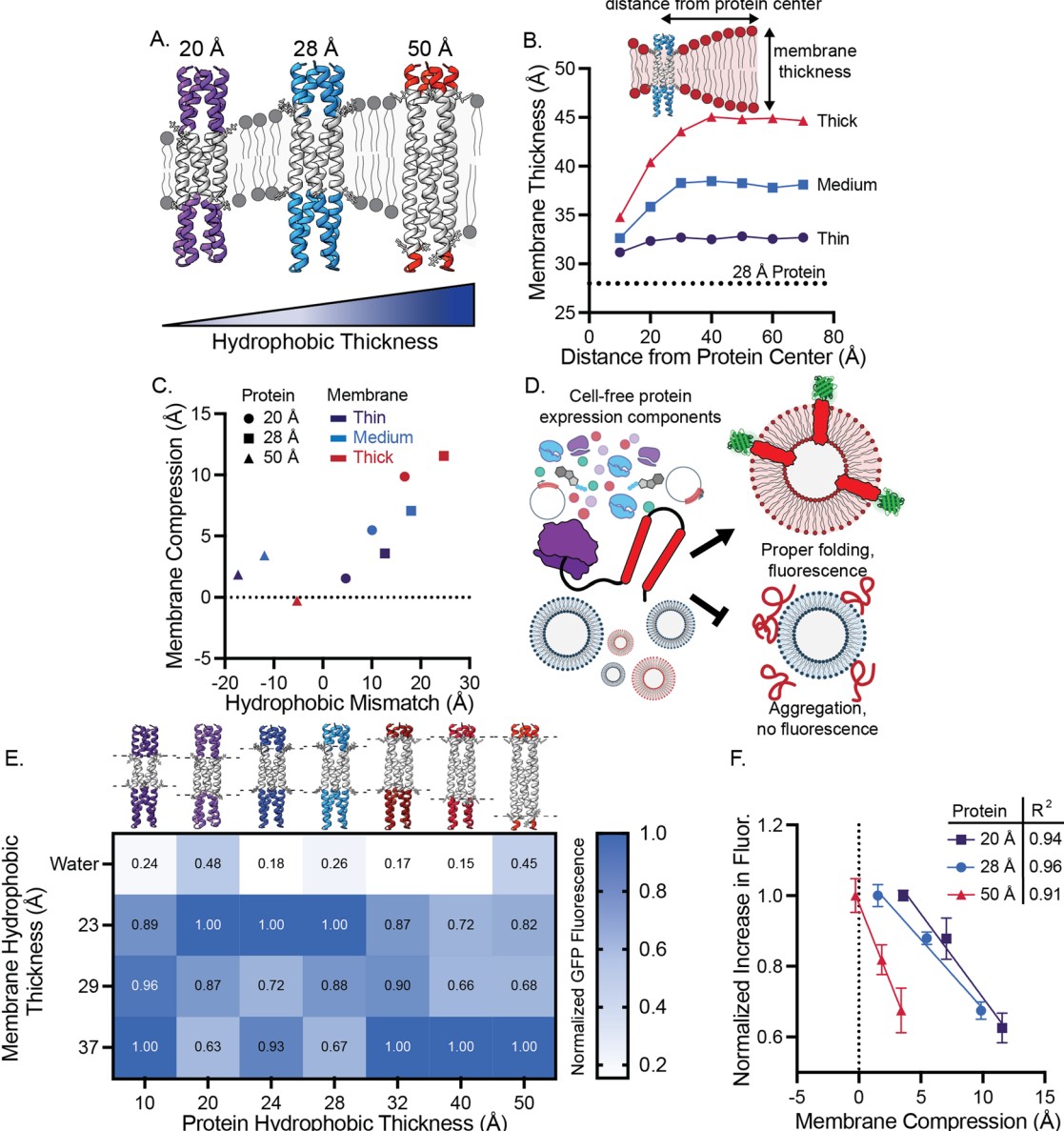

**Fig. 1 | Minimizing hydrophobic mismatch maximizes cell-free expression of membrane proteins into synthetic membranes. A** Interactions between de novo designed membrane proteins of varying hydrophobic thicknesses and synthetic membranes were explored. **B** The membrane thickness of a thin (DyPC), medium (DOPC), and thick (DGPC) membrane was determined as a function of distance from an inserted 28 Å protein using MD simulations. These simulations reveal that membranes must deform more to accommodate larger hydrophobic mismatch. The horizontal line represents the protein hydrophobic thickness. **C** Membrane compression is positively correlated with hydrophobic mismatch. Datapoints derived from simulations of the 20, 28, and 50 Å proteins by taking the difference in membrane thickness at the near the protein ($x = 10$ Å) and far away from the protein ($x = 70$ Å) for the three membrane compositions (thin (DyPC), medium (DOPC), thick (DGPC)). **D** The effect of hydrophobic mismatch on protein expression and folding in a cell-free protein synthesis systems was explored using mEGFP as a folding reporter. **E** Protein expression, as measured by increased GFP fluorescence, is maximized in hydrophobically matched membranes. Values represent the mean of three independent replicates, normalized to the maximum increase in fluorescence for each protein construct. Proteins were expressed in the presence of water (no membrane) and 23, 29, and 37 Å thick membranes (14:1 PC, 18:1 PC, and 22:1 PC, respectively). Membrane thicknesses for 14:1 PC, DOPC, and 22:1 PC were determined by Heberle et al. using small-angle x-ray scattering[31]. **F** Increase in GFP fluorescence is linearly correlated with membrane compression, as measured by MD simulation. Values represent the mean of three independent replicates, normalized to the maximum increase in fluorescence for each protein construct. Error bars represent the S. E. M.

results presented in Fig. 1C. We found that mEGFP expression for the 20, 28, and 50 Å correlated linearly with membrane compression (Fig. 1F). Together, these data demonstrate that membrane-protein hydrophobic mismatch inhibits membrane protein expression in cell-free systems.

To better understand the observed differences in protein expression, we next examined how transcription and translation were affected by hydrophobic mismatch. We measured transcription by

adding the DNA sequence for the malachite green aptamer immediately after the gene encoding the 50 Å protein. As the aptamer is transcribed, it binds to malachite green and dye fluorescence increases[35]. The presence of vesicles was found to inhibit transcription of the 50 Å protein; however, no significant differences in malachite green fluorescence were observed between the three lipid systems suggesting hydrophobic mismatch does not measurably affect transcription (Supplementary Fig. 6)[33]. We then set out to characterize how

translation is affected by membrane-protein hydrophobic mismatch. In separate work, we have demonstrated that the cell-free expression of several model membrane proteins results in the presence of more incomplete translation products and overall lower protein yield compared to soluble proteins expressed in the same cell-free systems. We developed a kinetic model to describe this phenomenon and attribute the reduced translation of membrane proteins relative to soluble proteins to increased aggregation of membrane protein, and believe a similar kinetic phenomenon affects protein expression as a function of hydrophobic mismatch[36]. To monitor the effect of hydrophobic mismatch on protein translation, we added an N-terminal FLAG tag to the 50 Å protein, allowing us to observe the formation of truncated protein products in addition to full-length protein products identified through the C-terminal mEGFP. Analyzing the total expression of the 50 Å protein in cell free reactions and protein associated with vesicles by western blot, we observed an increase in incomplete protein products relative to full-length protein as hydrophobic mismatch increases (Supplementary Fig. 7). The higher proportion of truncated protein products in the more hydrophobically mismatched system suggests that translation is affected by hydrophobic mismatch. We hypothesize this effect arises from the energy cost of deforming membranes to accommodate differences in hydrophobic mismatch, which reduces the probability of protein co-translational insertion and proper folding. Misfolded proteins likely capture nascent proteins

from the ribosome, thus reducing the rate of protein insertion and folding, and increasing the frequency of incomplete translation[36].

## Protein-lipid hydrophobic matching can be used to organize proteins between membranes and impart differentiated functionality

The differential expression and integration of membrane proteins into membranes of different thicknesses raised the possibility that this physical phenomenon could be used to enrich select populations of vesicles with a membrane protein in one pot. Based on the designs of our previous work, we created transmembrane pore proteins with a constitutively open 10 Å pore and with hydrophobic thicknesses ranging from 20 to 50 Å (Fig. 2A)[30]. To validate that pore proteins inserted into membranes, we expressed them in the presence of vesicles encapsulating calcein, a self-quenching dye (Fig. 2B). Upon expression of pore proteins, we observed calcein leakage and increased fluorescence[37]. We first confirmed that calcein leakage was specific to pore insertion (Supplementary Fig. 8), and then expressed the pores in the presence of vesicles with thick or thin membranes, encapsulating calcein. When normalized by protein expression, as determined by western blot, hydrophobically matched proteins released the most amount of calcein (Fig. 2C, D). This result demonstrates that reducing hydrophobic mismatch between the designed pore proteins and membranes maximizes the functional incorporation of this class of membrane proteins.

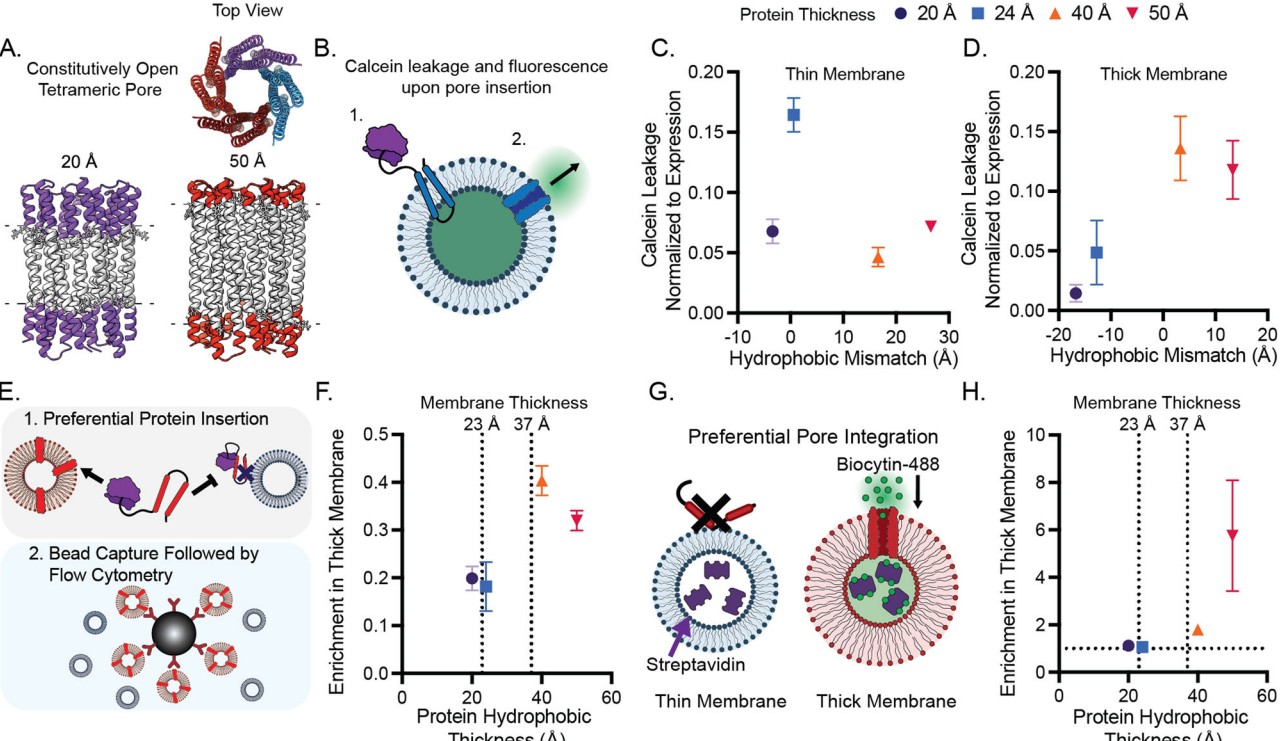

**Fig. 2 | Hydrophobic mismatch alone can organize cell-free expressed proteins between distinct membrane compartments. A** Constitutively open pore proteins of varying hydrophobic thicknesses were designed. **B** Proper folding and insertion of pore proteins was assessed via calcein leakage. Calcein leakage through de novo designed channel proteins is maximized when hydrophobic mismatch is minimized in both thin (14:1 PC, 23 Å), **C** and thick (22:1 PC, 37 Å), **D** membranes. Expression of protein in the presence of two populations of vesicles, followed by bead capture and flow cytometry enable the characterization of protein organization, **E**. As hydrophobic thickness of designed membrane channels is increased, the protein-mediated binding of thick membrane vesicles to beads increases relative to thin membrane vesicles, **F**. Enrichment in Thick membrane here is defined as the ratio of Rhodamine mean fluorescence intensity (22:1 PC) to Cy5.5 mean fluorescence intensity (14:1 PC). Values represent the mean of 5 independent replicates. Error

bars represent the S. E. M. **G**, To selectively deliver cargo in a mixed vesicle population, we expressed pore proteins in the presence of thick and thin membranes (14:1 and 22:1 PC, respectively), encapsulating streptavidin. Following protein expression, vesicles were incubated with AF488-biocytin, which could enter vesicles following pore integration. The amount of dye that was delivered to each population of vesicles was measured by flow cytometry. Vertical dotted lines in **F** and **H** correspond to membrane hydrophobic thickness. **H** Cargo delivery to thick and thin vesicles could be tuned by hydrophobic thickness of designed membrane pores. Enrichment in thick membrane is calculated as the ratio of Rhodamine labeled 22:1 PC vesicles positive for Biocytin-488 to Cy5.5 labeled 14:1 PC vesicles positive for Biocytin-488. All experiments were performed 3 times (**C**, **D**, **H**), error bars represent the S. E. M.

We next assessed the extent of protein expression and folding of a single protein (20, 24, 40, or 50 Å in hydrophobic thickness) into thick and thin membranes (22:1 PC, 37 Å and 23 Å, 14:1 PC respectively), when both membranes were present within one reaction. To evaluate differential integration, we developed a flow cytometry-based assay where each set of vesicles was labeled with an orthogonal lipid conjugated dye and each protein contained a C-terminal FLAG tag. Proteins were expressed in the presence of vesicles and were collected with anti-FLAG antibody-conjugated magnetic beads. Beads were analyzed by flow cytometry and read for colocalized vesicle fluorescence, which should occur by way of interactions of membrane-integrated proteins with the beads (Fig. 2E). We first validated this method by separately expressing the 40 Å hairpin protein into DOPC vesicles either labeled with 0.1 mol% 18:1 PE Rhodamine or 18:1 PE Cy5.5 dyes. We then mixed different ratios of these protein-integrated vesicles together, bound them to anti-FLAG beads, and analyzed their fluorescence via flow cytometry. We confirmed that shifts in fluorescence reflected the defined ratio of vesicles added to the beads (Supplementary Fig. 9). We then expressed either the 20, 24, 40, or 50 Å thick protein in the presence of a 1:1 mixture of thick and thin membranes (22:1 PC, 37 Å and 23 Å, 14:1 PC respectively). We added anti-FLAG beads to the vesicle mixture after protein expression and integration and measured the ratio of fluorescence from membrane dye in the thick membranes to membrane dye in the thin membranes that was colocalized to the beads. We found that as hydrophobic thickness of a protein increased, this ratio increased. Interestingly, this ratio was always below one for all proteins tested, suggesting all proteins have a preference for the thinner membrane, which would contradict the protein expression data we previously observed (Fig. 1). Alternatively, the FLAG tag on vesicle-integrated protein might be more accessible when in the thinner membrane, or there may be a propensity for the dye in the thinner membrane (18:1 PE-conjugated Cy5.5) to transfer to the thicker membrane, or vesicle instabilities may occur during the cell-free reaction leading to vesicle fusion. In any case, this result indicates that protein insertion into one population of membranes over another vesicle population can be biased by minimizing hydrophobic mismatch and as a result, the relative distribution of a protein between two populations of vesicles should be adjustable by changing the protein transmembrane domain length (Fig. 2F and Supplementary Fig. 9).

Accordingly, we explored the capacity of hydrophobic mismatch to assemble vesicles with a distinct functionality, in this case enhanced permeability, due to preferred integration of pore-forming membrane proteins. To assess functional integration of this protein, we measured protein-mediated entry of a biotinylated fluorophore (~1 kDa)[30]. We encapsulated streptavidin in the lumen of thick and thin membranes, each labeled with a distinct lipid-conjugated fluorescent dye. We expressed proteins of different hydrophobic thicknesses in the presence of both vesicles, purified away free streptavidin, and then incubated the vesicles with biocytin-conjugated AlexaFluor 488. Biocytin entry into vesicles, which should vary as a function of the number of functional pores in each vesicle membrane, could be monitored via vesicle-localized biocytin fluorescence since biocytin cannot leave the vesicles after it is bound to streptavidin in the vesicle lumen (Fig. 2G). Samples were then analyzed via flow cytometry and the ratio of thick and thin vesicles encapsulating AlexaFluor 488 were compared. As protein hydrophobic thickness increased, we observed an increased number of the thicker 40 Å vesicles contained captured dye relative to the thinner 23 Å vesicles. This result suggests that the population of vesicles to which biotin-AlexaFluor 488 was preferentially delivered could be tuned by biasing protein integration into hydrophobically matched vesicles membranes (Fig. 2H and Supplementary Fig. 10). Together, these data suggest that membrane compartments can be enriched with distinct protein content and therefore endowed with distinct function by modulating lipid-protein hydrophobic mismatch.

Our results highlight the capacity of protein-membrane hydrophobic mismatch alone to organize proteins between distinct membranes in vitro and suggest a route to engineer more complex membrane-based materials, such as differentiated-nested vesicles or synthetic organelles[38].

## Hydrophobic mismatch coupled with phase-separating lipid mixtures controls protein-protein interactions within a single membrane

The lateral organization of membrane proteins in a single membrane is important to control protein-protein and protein-lipid interactions and subsequent signaling activity[13,39,40]. This organization arises due to different lipid-lipid, lipid-protein, and cytoskeletal interactions. While the functional relationship between protein organization and signaling has been explored in cellular contexts[40], it has not successfully been recapitulated in vitro. Demonstrating how membrane organization can be leveraged to control protein-lipid and protein-protein interactions, and subsequent signaling, in synthetic systems is critical to understanding the molecular and physical origin of these interactions. Here, we explore how de novo designed alpha helical proteins can be organized in synthetic membranes based on hydrophobic mismatch alone and demonstrate that this organization can be used to control the function of an enzyme, luciferase. In doing so, we not only uncover the extent to which protein and lipid-driven organization may enable protein organization, interactions, and subsequent activity, but also provide a route to design more complex sensing and signaling modalities within membrane-based materials.

Previous work has explored how lipid composition and hydrophobic mismatch impact the organization of lipids and proteins within membranes in vitro[11,32,41–45]. It has been demonstrated that peptides can be laterally organized through membrane ordering[45] and electrostatic interactions[44] in unsaturated lipid systems and that beta-barrel proteins can associate with liquid-ordered lipid phases through the modulation of protein hydrophobic thickness[46]. Further, it has been shown that proteins preferentially interact with hydrophobically matched membrane components in non-phase separated supported lipid bilayers[47]. However, contradicting phase behavior of proteins in cellular, in silico, and synthetic membranes has been noted[48,49], likely due to the use of microdomain-forming lipid mixtures in synthetic lipid systems, which are more ordered than biological membranes, hindering protein association with ordered lipid phases. We hypothesized that by designing membranes just above a lipid de-mixing transition, like biological membranes[50], we could observe induction of lipid domains induced by local changes in curvature or hydrophobic thickness around membrane components, such as proteins[51–53] and experimentally demonstrate the segregation of two types of proteins to distinct regions of a membrane.

To examine the ability of hydrophobic mismatch to affect protein location and protein-protein interactions in a single membrane, we first characterized how single proteins co-localize with lipid components based on hydrophobic mismatch. We prepared membranes with a shorter unsaturated lipid, 14:1 PC, and a thicker saturated lipid, DPPC (16:0 PC), and cholesterol. This combination of lipids is prone to phase separation and at different lipid ratios can form homogenous and phase separated membranes[54]. We prepared a membrane composed of 42.5 mol% 14:1 PC/27.5 mol% DPPC/30 mol% Chol as this lies just above a de-mixing transition and does not form microdomains at room temperature[9,48,49,55] (Supplementary Fig. 11). We first simulated membrane interactions with thin (20 Å) and thick (50 Å) proteins using coarse-grained MD simulations of lipid composition comparable to the experimental system (42 mol% DyPC PC/28 mol% DPPC/30 mol% Chol). We observed that insertion of membrane proteins into an initially homogenous lipid mixture induced lipid reorganization. Distinct lipid-protein domains formed: a domain rich in thinner, unsaturated lipid (DyPC) appeared around the 20 Å protein and a domain rich

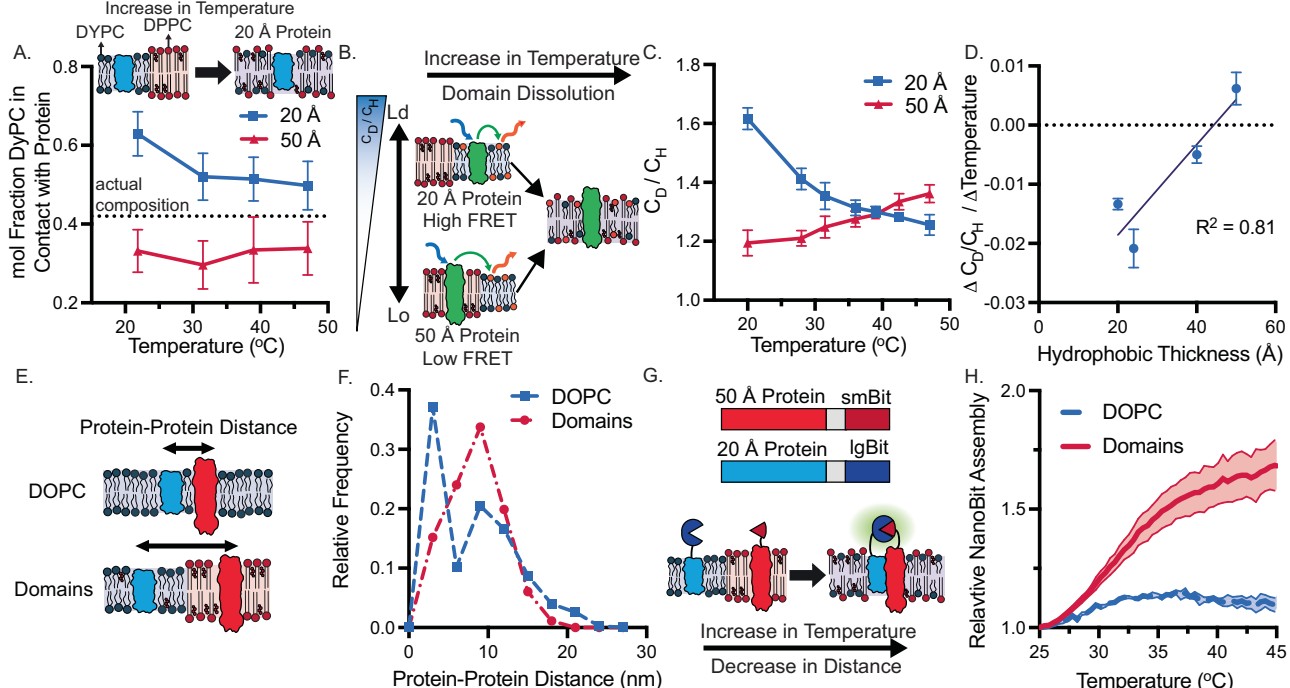

**Fig. 3 | Lipid-protein hydrophobic mismatch can dynamically tune protein–protein interactions. A** The 20 and 50 Å thick proteins were simulated in 42 mol% DyPC/28 mol% DPPC/30 mol% Cholesterol membranes. mol Fraction of DyPC in contact with the protein was determined by considering the time-averaged membrane composition of lipids in the first shell around each protein. These simulations indicate the 20 Å hairpin proteins interact more with the shorter lipid, DyPC, compared to the 50 Å protein. As temperature increases, the protein-DyPC contacts shift towards the average membrane composition. The dotted line indicates the actual composition of DyPC, 42 mol%. **B** Lipid-protein FRET between C-terminal AlexaFluor 488-SNAP tag and Rhodamine conjugated lipids enable the evaluation of protein organization within synthetic membranes. Experimental data in panels **C**, **D** are reported in domain forming membranes (42.5% 14:1 PC/27.5% DPPC/30% Chol) and compared to homogenous membranes (100% DOPC) in panels **F**–**D**. **C** 20 Å and 50 Å proteins associate differently with Rhodamine conjugated lipids (18:1 PC), as reported by $C_D/C_H$. As temperature increases, the membrane becomes more fluid enabling lipid mixing and convergence of the two $C_D/C_H$ curves. **D** Total change in $C_D/C_H$ from 20 to 45 °C correlates with protein hydrophobic thickness. **E**, **F** Protein-protein distance can be modulated by lipid composition of synthetic membranes. In homogenous DOPC membranes, 20 Å and 50 Å proteins can be close to one another, however in phase separating lipid mixtures proteins remain farther from one another as predicted by MD simulations. **G**, **H** Lipid domain forming membranes separate integrated 20 Å and 50 Å proteins at room temperature. Upon heating, to enable protein and lipid mixing, split luciferase reconstitution and subsequent luminescence is higher in the domain forming lipid mixture, compared to DOPC. All experiments were performed three times, error bars represent the S. E. M.

in the thicker, saturated lipid (DPPC) appeared around the 50 Å protein over time (Fig. 3A, Supplementary Fig. 12, and Supplementary Move 1). We then studied the effect of increasing temperature, a means to dissolve lipid-protein domains as saturated and unsaturated lipids become more miscible at elevated temperatures[54]. As the temperature increased, protein-lipid contacts between the 20 Å protein and DyPC converged toward the average composition of the membrane (Fig. 3A).

We then experimentally assessed how individual proteins were organized in our membranes via lipid-protein FRET (fluorescence resonance energy transfer). To accomplish this, we added rhodamine dye conjugated to 18:1 PE into our membranes, which localizes with shorter, unsaturated lipids, and fused a C-terminal SNAP tag to each protein, allowing conjugation of AlexaFluor 488. Using FRET between SNAP Alexa Fluor 488 and the lipid-conjugated rhodamine dye, we could calculate the local concentration of rhodamine around the protein in domain-forming membranes (42.5% 14:1 PC/27.5% DPPC/30% Chol) compared to homogenous membranes (100% DOPC), represented as $C_D/C_H$ (see Methods). Using this metric, $C_D/C_H$ will be higher when proteins and dye partition to the same lipid domain and low when they partition to separate domains (Supplementary Fig. 13 and Fig. 3B)[46]. We measured $C_D/C_H$ values of the 20 and 50 Å protein over a range of temperatures. At room temperature, the 20 Å protein had a higher $C_D/C_H$, indicating that it resides in the dye containing, thinner, and more unsaturated 14:1 PC rich phase. Upon increasing temperature to dissolve domains, we observed that the average

distance between the unsaturated lipid dye and thinner protein increased. Conversely, the 50 Å protein had lower $C_D/C_H$ values at lower temperatures that increased with increasing temperature. This result suggests that the 50 Å protein interacts more with the rhodamine-labeled lipid when present in a ternary composition membrane than when present in the homogenous one. The membranes we generated for experimental studies are not phase separated in the traditional sense with a stable liquid ordered and disordered phase, especially because we conducted these studies close to the melting transition temperature of the tertiary lipid composition. We do not observe micro domains in GUVs and in simulations in the absence of proteins. In contrast, local phase separation or an increase in lipid organization is only observed when proteins are present in the membrane. Nevertheless, the increase in $C_D/C_H$ values as temperature is increased suggests the larger 50 Å protein moves from a state where its interactions with the dye poor, thicker, and more saturated DPPC rich lipid phase decrease and the protein interacts more with the dye-rich, thinner and more unsaturated 14:1 PC lipids (Fig. 3C). The FRET data obtained as a function of temperature for the 20 Å protein mirrors our simulation data in Fig. 3A[6,7]. However, domains around the 50 Å protein in the simulation appear to be more stable than in the synthetic membrane experiments, likely due to different parameters between the MARTINI model and experimental conditions[56,57]. Further, changes in $C_D/C_H$ for the 20, 24, 40, and 50 Å hairpin proteins upon increases in temperature are dependent on protein hydrophobic thickness: as

protein thickness increased relative to the membrane, there were greater shifts in the local concentration of disordered lipids around the protein (Fig. 3D and Supplementary Fig. 14). These results demonstrate lipid composition around a protein can be tuned by hydrophobic mismatch. Together the simulations and FRET data suggest that hydrophobic mismatch coupled with lipid domain formation can be leveraged to organize proteins into distinct regions within synthetic membranes.

As a next step, we wondered if we could modulate protein-protein interactions by localizing two proteins of different transmembrane length to separate lipid domains. Cells control membrane protein interactions using lipid domains, particularly in immune signaling[39], and mimicking this strategy could offer substantial advantages in the design of transmembrane signaling transduction in membrane-based technologies[15,20]. We first investigated how two proteins interact with one another by performing MD simulations of the 20 Å and 50 Å hairpin protein in a homogenous, single component (DOPC) and heterogenous, ternary membranes (42.5 mol% DyPC/27.5 mol% DPPC/30 mol% Chol) (Fig. 3E). These simulations indicated that proteins were able to be close to one another in homogenous membranes but remained separated in our mixed, phase segregated membrane (Fig. 3F, Supplementary Figs. 12 and 15, and Supplementary Movie 1).

We then investigated if this behavior could be recapitulated experimentally and used to control assembly of a split protein. We fused rapamycin inducible-dimerizing domains and NanoBit, a split nano luciferase[58], to the C terminus of the 20 and 50 Å proteins. To assess how lipid domains affected protein compartmentalization and subsequent NanoBit assembly, we co-expressed smBit and lgBit fused to the 20 Å and 50 Å protein, respectively, into homogenous or phase-separated membranes. We observed how luminescence changed following the addition of rapamycin[15], which chemically induces protein dimerization and subsequent luciferase assembly. Upon addition of rapamycin, we saw an increase in luminescence when proteins were in homogenous membranes; however, we observed minimal increases in phase-separated membranes, suggesting that proteins were less able to dimerize and assemble luciferase due to segregation into different lipid domains (Supplementary Fig. 16). We then performed a temperature ramp on these systems to dissolve lipid domains (Fig. 3G). In phase-separated systems, we observed an increase in luminescence with temperature when lgBit and smBit are fused to the hetero-pair of 20 and 50 Å protein, respectively, relative to when smBit and lgBit are both conjugated to homo-pairs of either the 20 or 50 Å proteins. Importantly, only a slight increase in luminescence is observed with increase in temperature when proteins are in homogenous membranes, indicating the proteins are more evenly distributed in the homogenous membrane and so less sensitive to increases in lipid mixing upon temperature increases (Fig. 3H and Supplementary Fig. 17). Combined, the simulation and experimental data suggest lipid-lipid and lipid-protein interactions can together be harnessed to modulate protein interactions in a single membrane. Such control should improve the specificity and off target, ligand-independent activation of engineered receptor systems in both cellular and synthetic membranes[24].

## Discussion

Here, we report on the capacity of hydrophobic mismatch between membranes and membrane proteins to drive changes in both membrane protein synthesis and protein location. Using cell-free systems to recapitulate membrane protein synthesis and folding outside of the cellular environment, we demonstrate that proteins not only cotranslationally insert and fold better into hydrophobically matched membranes, but also that increases in mismatch reduce the yield of protein that is synthesized. Once in a mixed composition membrane that is capable of phase segregation, hydrophobic mismatch between lipids and proteins can drive reorganization to segregate lipids and proteins

of similar length. Capitalizing on these inter-membrane and intra-membrane sorting mechanisms, we demonstrate: (1) the one-pot assembly of membranes with preferential protein incorporation and corresponding function based on hydrophobic mismatch alone and (2) control over the interactions of de novo designed proteins in a single membrane. These in vitro results underscore the importance of hydrophobic matching for proper protein folding in biological systems and highlight how such physical features may be leveraged to enhance synthetic membrane-based materials. Specifically, leveraging physio-chemical interactions of lipids and proteins between distinct membranes and within the hydrophobic region of single membranes will enable the design of membrane-based materials with enhanced transmembrane signaling[15,20,24,28,59] and more effective and specific engagement with cells[23,27], leading to more effective therapeutics.

This work is a major step towards utilizing de novo protein design, MD simulation, and membrane augmented cell-free systems to characterize complex biophysical phenomena, from the bottom up. Here, we show that protein hydrophobic mismatch alone can be used to enrich specific membranes with protein content and modulate protein-protein interactions. Beyond the systems explored in this study, we envision that leveraging synthetic membranes and de novo designed proteins will allow us to characterize the impact of a variety of membrane biophysical properties in a systematic, and controlled manner, and explore space not observed in natural systems. Altogether, coupling de novo membrane protein design with molecular dynamics and cell-free systems highlights a powerful workflow which will enable a better understanding of membrane protein biophysics and inform the future design of membrane-based materials.

## Methods

### Protein design

We designed transmembrane proteins with different transmembrane spans (with a range of 20–50 Å) by resurfacing the outside of the de novo designer transmembrane proteins with patterned hydrophobic residues and adding RK- and YW-rings at the intracellular and extra-cellular boundary region, respectively. Briefly, hydrophobic residues are designed based on amino acid propensity in the membrane, replacing all polar residues exposed to the membrane. The design models of TMHC2 and TMH4C4 were used as the starting model. Amino acid sequences of transmembrane proteins can be found in Supplementary Table 2.

### Coarse-grained simulations

Coarse-grained molecular dynamics simulations were conducted using the MARTINI force field (v2.2) using GROMACS (2020.1). Simulations were performed using semi-isotropic pressure coupling to yield laterally tensionless membranes using the "martini straight" parameters, a timestep of 30 fs, the neighbor list was updated using the Verlet neighbor search, with the neighbor list length being automatically determined. LJ and Coulomb potentials and forces were cut off at 1.1 nm, with the potentials shifted to zero at the cut off. The neighbor list was updated every 20 steps. Velocity rescale and Parrinello–Rahman coupling schemes were used with coupling parameters of 1.0 and 12.0 $ps^{-1}$ (ref. 60). The secondary structure of simulated protein was fixed in the simulations by an elastic network parametrized from the predicted protein structure[61,62]. The protein constructs and membranes of varying lipid compositions were assembled using insane.py[63] and initially equilibrated for a minimum of 10 ns, productions runs were 6 μs with three replicates and sampled every 1 ns. If not indicated otherwise, the simulation was conducted at 295° K. Trajectories were analyzed using MDAnalysis version 0.20.1[64]. The used parameters, coarse-grained model and equilibration time were chosen and validated based on previous studies and were found to be independent from the initial starting configuration (https://doi.org/10.1021/jp4000686, https://doi.org/10.1016/j.bpj.2018.03.020).

The MARTINI model and used timescales are well suited to capture the elastic membrane deformation studied (ref. 52). Type of analysis and simulation size varying between systems: Data in panel Fig. 1B, C was obtained for single component membranes with 216 MARTINI DyPC, DOPC or DGPC lipids per leaflet and 7587 water molecules. Resulting in a box size of about $12 \times 12 \times 10$ nm$^3$ and a total of 13226 coarse-grained beads. The position in normal direction to the membrane of the PO4 bead (representing the phosphate headgroup) was analyzed around a single-centered protein construct for both membrane leaflets. Then PO4 positions were binned by the radial distance from the protein center with 1 nm bin width. The difference between the PO4 position for each leaflet bin then determined the membrane thickness shown as an average over the whole trajectory. Membrane compression was calculated by subtracting the membrane thickness at 70 Å from membrane thickness at 10 Å from the protein center and is represented by the following equation:

$$Membrane\ compression = Thickness_{x=70Å} - Thickness_{x=10Å} \quad (1)$$

In simulations, DyPC (di-C12:1-C14:1 PC) was used in place of 14: PC (di-C14:1-C14:1) and DGPC (di-C20:1-C22:1) was used instead of 22:1 PC (di-C22:1-C22:1) due to the availability of Martini lipids. We believe that these differences are small and that the simulations are still reflective of the experimental system.

For simulations shown in Fig. 3A, membranes were composed of 138 DYPC, 92 DPPC, and 99 cholesterol molecules per leaflet and 17101 water molecules. Resulting in a box size of about $20 \times 20 \times 8.3$ nm$^3$ and 33720 coarse-grained beads. In a radial selection around the protein center of mass, corresponding to the first layer of surrounding lipid molecules, individual lipid types were determined. The time average of detected lipids, then determined average membrane composition around the protein center at varying temperatures. For Fig. 3F the same DPPC:DyPC:cholesterol membranes as above were compared to membranes with 326 DOPC lipids per leaflet. Both membranes contained two copies of two different protein constructs. The distributions of center of mass protein-protein distances were determined for the two membrane compositions.

All simulations were performed in at least triplicate. Systems simulated and time simulated can be found below in Table 1.

## Gene assembly and cloning

Genes listed in Supplementary Table 4 were ordered as gene blocks and cloned into a high copy plasmid used in previous work[65]. gBlocks and primers were ordered from Integrated DNA technologies. Different fusion proteins were generated using standard restriction enzyme cloning techniques using Phusion DNA polymerase and restriction enzymes from Thermo Fisher. Pore proteins were toxic and prone to mutation and thus were not cloned into plasmids. Protein pores were ordered as gene blocks with elements required for gene transcription and translation (T7 promoter and terminator, ribosome binding site) and were amplified via PCR.

## Vesicle preparation

Throughout this study, vesicles were prepared via the thin film hydration method. Briefly, lipid was deposited into a glass vial and dried with a stream of nitrogen and placed under vacuum for 3 h. Films were then rehydrated in Milli-Q water and heated at 60 °C for a minimum of 3 h, and up to overnight. Vesicles were then briefly vortexed and extruded 21x through a 100 nm polycarbonate filter. All lipids used in this study (1,2-dioleoyl-sn-glycero-3-phosphocholine (DOPC), 1,2-dimyristoleoyl-sn-glycero-3-phosphocholine (14:1 PC), 1,2-dierucoyl-sn-glycero-3-phosphocholine (22:1 PC), Cholesterol, 1,2-dipalmitoyl-sn-glycero-3-phosphocholine (DPPC), 1,2-dioleoyl-sn-glycero-3-phosphoethanolamine-N-(7-nitro-2-1,3-benzoxadiazol-4-yl)1,2-dioleoyl-snglycero-3-phosphoetha-nolamine-N-(lissamine rhodamine B sulfonyl) (18:1 Rhodamine PE), and

**Table 1 | Summary of systems simulated**

| Measurement | Figure | Protein | Membrane composition | Simulation time [µs] × replicates | Total simulation time per lipid composition [µs] |
|---|---|---|---|---|---|
| Membrane thickness and compression | Fig. 1B, C, F, Supplementary Fig. 1 | Single protein (20, 28, 50 Å proteins) | DOPC, DYPC, or DGPC | 6 µs × 3 | 18 µs |
| Lipid-protein interactions | Fig. 3A | Single protein (20 or 50 Å proteins) | 42 DYPC:28 DPPC: 30 CHOL | 6 µs × 3 | 18 µs |
| Protein-protein interactions | Supplementary Fig. 15 | 20 Å – 20 Å proteins | DOPC or 42 DYPC:28 DPPC: 30 CHOL | 6 µs × 3 | 18 µs |
| Protein-protein interactions | Supplementary Fig. 15 | 50 Å – 50 Å proteins | DOPC or 42 DYPC:28 DPPC: 30 CHOL | 6 µs × 3 | 18 µs |
| Protein-protein interactions | Fig. 3F, Supplementary Figs. 12 and 15 | 20 Å – 50 Å proteins | DOPC or 42 DYPC:28 DPPC: 30 CHOL | 6 µs × 9 | 54 µs |

1,2-dioleoyl-sn-glycero-3- phosphoethanolamine-N-(Cyanine 5.5) (Cy 5.5 PE)) 1,2-dipalmitoyl-sn-glycero-3-phosphoethanolamine-N-(7-nitro-2-1,3-benzoxadiazol-4-yl) (ammonium salt) (16:0 NBD PE), 1,2-dioleoyl-sn-glycero-3-phosphoethanolamine-N-(7-nitro-2-1,3-benzoxadiazol-4-yl) (ammonium salt) (18:1 NBD PE) were purchased from Avanti Polar Lipids. A table summarizing the physical features of lipids used in this study can be found in Supplementary Table 1, and the lipid compositions used throughout the study can be found in Supplementary Table 3. Reported membrane thicknesses for 14:1 PC, DOPC, and 22:1 PC were determined by Heberle et al. using small-angle x-ray scattering[31].

### Analysis of folding and insertion of cell-free expressed proteins into synthetic membranes

Protein expression was performed with the PURExpress In Vitro Protein Synthesis kit (E6800, NEB) according to the manufacturer's instructions. 30 μL reactions were assembled with a final concentration of 10 mM of lipid and 3.3 nM plasmid. Reactions were allowed to progress at 37 °C for 3 h. GFP folding and fluorescence was monitored using a Molecular Devices Spectra Max i3 plate reader (ex 480 nm, em. 507 nm). Increase in GFP fluorescence was then calculated by subtracting the fluorescence at t = 0 from the fluorescence at $t = 3$ h.

Protein expression was measured via western blot. Cell-free protein synthesis reactions were spun at $20,000 \times g$ for 10 min to pellet and remove uninserted protein. The supernatant was collected and run on a 12% Mini-PROTEAN TGX Precast Protein Gel (Bio-Rad) for all experiments, except the truncation experiments. For truncation experiments, samples were run on a 16.5% Tricine Mini-PROTEAN Precast Protein Gel to enhance the separation of smaller protein products. To observe truncation products associated with vesicles, 1 mol% Biotinyl Cap PE was included in each membrane. 3.5 μL of cell-free reaction mixture was mixed with 0.1 mg of Pierce Streptavidin beads, incubated at room temperature for 1 h, and washed at least twice until supernatant A280 reached <0.05. Resuspended bead mixtures were mixed with loading dye and loaded onto a Tricine Mini-PROTEAN Precast Protein Gel. For all western blots, wet transfer was performed onto a PVDF membrane (Bio-Rad) for 45 min at 100 V. Membranes were then blocked for an hour at room temperature in 5% milk in TBST (pH 7.6: 50 mM Tris, 150 mM NaCl, HCl to pH 7.6, 0.1% Tween) and incubated for 1 h at room temperature or overnight at 4 °C with primary solution (anti-GFP (Abcam, ab-290) for full-length expression and anti-Flag (Sigma F1804) for truncation experiments, diluted 1:1000 in 5% milk in TBST). Primary antibody solution was decanted, and the membrane was washed three times for 5 min in TBST and then incubated in secondary solution at room temperature for 1 h (HRP-anti-Mouse (CST 7076) diluted 1:3000 in 5% milk in TBST). Membranes were then washed in TBST and incubated with Clarity Western ECL Substrate (Bio-Rad) for 5 min. Membranes were then imaged in an Azure.

Insertion of proteins into synthetic membranes was further characterized by size exclusion chromatography. Briefly, soluble mEGFP and the 28 Å protein fused to mEGFP were expressed using PURExpress (NEB) in the presence of either 10 mM 14:1 PC, DOPC, or 22:1 PC with 0.1 mol% 18:1 PE Cy5.5. Reactions were then purified via size exclusion column packed with Sepharose 4B (45 – 165 mm bead diameter) to separated protein integrated into vesicles from free protein. Elution fractions were read on a plate reader (Molecular Devices Spectra Max i3) for mEGFP (ex. 480 nm/em. 505 nm) and Cy5.5 (ex. 678 nm/em. 707 nm) fluorescence.

### Preparation of giant unilamellar vesicles

Giant, micron sized, vesicles were prepared via electroformation using the Nanion Vesicle Prep Pro (Nanion Technologies) standard vesicle preparation protocol. To visualize protein, proteins were expressed into liposomes containing 0.1 mol% 18:1 PE Cy5.5. Following expression, liposomes were diluted to 1 mM and 10 μL were deposited onto

indium tin oxide slides and allowed to dry under vacuum for 30 min. Samples were then rehydrated with 290 mOsm sucrose. To visualize domains, 10 mM mixtures of lipid in chloroform were prepared with 0.1 mol% Rhod-PE. 10 μL of each solution was then drop-casted onto indium tin oxide slides and placed under vacuum for 20 min to eliminate solvent and rehydrated with 290 mOsm sucrose. GUVs were observed under a Nikon confocal microscope. Glass-bottomed Lab-Tek II microscope chambers (Thermo Fischer) were used to image GUVs. 200 μL of 1% bovine serum albumin in PBS was placed into each chamber and allowed to sit for 30 min. Each well was then washed with 290 mOsm PBS and 1 mL of 1 mM of GUVs were added to 250 mL of PBS and allowed to settle in each chamber. A 20x objective was used to visualize vesicles. Images were analyzed using NIS software.

### Calcein leakage

Vesicles were rehydrated with 50 mM Calcein in 10 mM HEPES. Calcein vesicles were purified using a size exclusion column packed with Sepharose 4B (45–165 mm bead diameter) immediately before experimentation. PURExpress reactions were then assembled and calcein leakage was read (ex. 480 nm/em. 520 nm) on the plate reader (Molecular Devices Spectra Max i3) for 3 h at 37 °C. 1% Triton-X was then added to achieve a maximum dequenching of calcein, which served as the fluorescence intensity for 100% mixing. Percent content mixing was calculated using the following equation:

$$\%Calcein\,Release = 100 \cdot \frac{I_{t=3\,hr} - I_{t=0\,hr}}{I_{triton} - I_{t=0\,hr}} \qquad (2)$$

where $I_{t=0\,hr}$ is the initial fluorescence intensity, $I_{t=3\,hr}$ is the fluorescence intensity at 3 h, and $I_{triton}$ is the fluorescence intensity after the addition of Triton-X. To determine the relative calcein release per protein, western blots were performed on samples. Calcein release values were then divided by total protein intensity for each sample to calculate the calcein release relative to protein expression.

### Assessing protein sorting between distinct compartments via immunoprecipitation

100 nm 14:1 and 22:1 PC vesicles were prepared as outlined above with 0.1 mol% 18:1 PC Cy5.5 and 18:1 PC Rhodamine respectively. PURExpress reactions were assembled with 3.3 nM plasmid encoding either the 20, 24, 40, or 50 Å pore protein and 5 mM each of 14:1 and 22:1 PC lipid. Reactions were allowed to progress at 37 °C for 3 h. Samples were then incubated with Pacific-blue anti-FLAG antibody (CST, D6W5B) conjugated protein A/G beads for 1 h at room temperature. Samples were washed 3 times and then analyzed via flow cytometry. Beads were gated for size (only larger beads were selected to eliminate unbound vesicles) and anti-FLAG antibody (405 nm excitation, 450/50 nm emission). Beads were analyzed for Rhodamine (550 nm excitation, 582/15 nm emission) and Cy5.5 (640 nm excitation, 730/45 nm emission with 685 longpass filter). At least 10,000 events were recorded, and beads were re-gated in FlowJo (TreeStar). Enrichment in the thick membrane was calculated as follows:

$$\begin{aligned} Enrichment\,&in\,thick\,membrane \\ &= MFI(Rhodamine, 22:1PC)/MFI(Cy5.5, 14:1PC) \end{aligned} \qquad (3)$$

### Analyzing differential pore activity

14:1 and 22:1 PC vesicles were prepared as outlined above with 0.1 mol% 18:1 PC Cy5.5 and 18:1 PC Rhodamine respectively. Lipid films were rehydrated with 5 μM streptavidin and extruded to 1 μm. PURExpress reactions were assembled with 3.3 nM plasmid encoding either the 20, 24, 40, or 50 Å pore protein and 5 mM each of 14:1 and 22:1 PC lipid and incubated at 37 °C for 3 h. Reactions were then purified via size exclusion chromatography to purify away unencapsulated

streptavidin. Vesicles were incubated with 1 μM biocytin conjugated Alexa Fluor 488 for 24 h. Samples were diluted to a lipid concentration of 1 μM in PBS and analyzed via flow cytometry on a BD LSR Fortessa Special Order Research Product (Robert H. Lurie Cancer Center Flow Cytometry Core). Alexa Fluor 488 was excited with a 488 nm laser and captured with a 505 nm long pass filter and a 530/30 nm bandpass filter, Rhodamine was excited with a 552 nm laser and captured with a 582/15 nm bandpass filter, and Cy5.5 was excited with a 640 nm laser and captured with a 685 nm longpass filter and a 730/45 nm bandpass filter. Events on the cytometer were thresholded on the presence of either Rhodamine or Cy5.5 detection to identify vesicles, and approximately 100,000 events were captured per reaction. Data was analyzed in FlowJo v10.8 and spectrally compensated. Samples were gated using curly quad gating of Rhodamine versus Cy5.5 to isolate single-dye positive events and thus restrict analysis to only thin or thick membrane vesicles (Supplementary Fig. 10). Curly quad gating (rather than quad gating) was necessary to account for photon counting and measurement error at high laser settings[66]. Single lipid-dye positive events were then gated for Alexa Fluor 488 using samples containing vesicles but no Alexa Fluor 488. The percent of thin and thick vesicles identified as Alexa Fluor 488 positive and the mean fluorescent intensity (MFI) of each vesicle population was determined and analyzed. Vesicles that were prepared as above but that did not have a protein pore (i.e., the PURExpress reaction did not have pore-encoding DNA) was used to determine background. This background signal (e.g., percent positive vesicles or MFI) was subtracted as part of the data analysis process.

### Lipid-Protein FRET experiments
Vesicles composed of DOPC or 42.5 mol% 14:1 PC/27.5 mol% DPPC/ 30 mol% cholesterol were prepared with 0.1 mol% 18:1 PC Rhodamine as outlined above. PURExpress reactions were prepared with 10 mM vesicles and 3.3 nM plasmid encoding the 20, 24, 40, or 50 Å hairpin proteins with a C-terminal SNAP tag. Reactions were performed at 37 °C for 3 h. Samples were then incubated with 10 μM Alexa Fluor-SNAP substrate (NEB) for 30 min at 37 °C. Vesicles were purified away from free SNAP substrate via size exclusion chromatography using Sepharose 4B (45–165 mm bead diameter) (Sigma Aldrich). Vesicles were collected and FRET was measured using an Agilent Cary Eclipse Fluorescence Spectrophotometer by exciting the samples at 488 nm and recording the emission at 520 and 590 nm. Fluorescence measurements were recorded at temperatures ranging from 25 to 47 °C. Vesicle samples were then treated with trypsin and 0.1% Triton X to disrupt vesicles and SNAP conjugated dye.

Relative FRET, noted here as $C_D/C_H$, was calculated using the following equation:

$$C_D/C_H = \ln\left(\frac{F}{F_o}\right)_D \Big/ \ln\left(\frac{F}{F_o}\right)_H \tag{4}$$

where F is the fluorescent intensity of donor in the presence of acceptor, $F_o$ is the fluorescent intensity of donor after the addition of trypsin and Triton-X. D denotes samples with domain forming membrane and H denotes samples with homogenous membranes. With this convention, $C_D/C_H$ will be high if Rhodamine (acceptor) and protein (donor) partition into the same lipid domain and low if they are segregated into different lipid domain[46].

### NanoBit experiments
Vesicles composed of DOPC or 42.5 mol% 14:1 PC/27.5 mol% DPPC/ 30 mol% Cholesterol were prepared as outlined above and extruded to 100 nm. PURExpress reactions were assembled with 1.7 nM of each DNA construct: 20 Å Hairpin/ 50 Å Hairpin, 20 Å Hairpin/ 20 Å Hairpin, 50 Å Hairpin/ 50 Å Hairpin. Reactions were allowed to progress for 3 h at 37 °C.

For rapamycin experiments, cell-free reactions were split into two and either rapamycin in DMSO or DMSO only was added to protein-incorporated vesicles at a final concentration of 30 nM (or a DMSO mol fraction of 1 mol lipid: 0.0015 mol rapamycin). Samples were incubated for 2 h at room temperature. NanoBiT reactions were setup using the Promega Nano-Glo Live Cell Assay System following the Technical Manual with minor modifications. Cell free reactions were diluted 1:4 in 1x PBS and the Nano-Glo Substrate was used at a 50x final dilution of the stock. Luminescence was read using a Molecular Devices Spectra Max i3 plate reader at room temperature for 10 min. To ensure the ratios of NanoBit to Substrate were in optimal range, luminescence was checked to be constant over the 10-min read. Rapamycin-induced luminescence was then calculated as:

$$Rapamycin\ Induced\ Lum. = Luminesence_{+Rap}/Luminesence_{-Rap} \tag{5}$$

Where luminescence$_{+Rap}$ is the measured luminescence in the presence of rapamycin and luminescence$_{-Rap}$ is the measured luminescence in the presence of DMSO only.

To characterize protein-protein interactions with increasing temperature, the luminescence of samples was then recorded at varying temperatures from room temperature to 45 °C. Relative NanoBit assembly was then calculated as:

$$Relative\ NanoBit\ Assembly = Lum._{20\text{Å}-50\text{Å}}/0.5 \\ * (Lum_{20\text{Å}-20\text{Å}} + Lum_{50\text{Å}-50\text{Å}}) \tag{6}$$

where Lum.$_{20\text{Å}-50\text{Å}}$ is the luminescence of samples with 20 Å and 50 Å hairpin proteins, Lum.$_{20\text{Å}-20\text{Å}}$ is the luminescence of samples with 20 Å and 20 Å hairpin proteins, and Lum.$_{50\text{Å}-50\text{Å}}$ is the luminescence of samples with 50 Å and 50 Å hairpin proteins. Luminesce values were then normalized to the luminesce value at room temperature. Dividing by the average of NanoBit fused to proteins of the same length allows for the increase in Nanobit assembly due to increases in lipid and protein mixing as systems with the same TMDs should reside in the same lipid domains. Furthermore, this normalization accounts for luminescence differences due to temperature.

### Reporting summary
Further information on research design is available in the Nature Portfolio Reporting Summary linked to this article.

## Data availability
All data can be found in the manuscript and supplementary files. DNA sequences encoding proteins and their descriptions, plasmids used in each experiment can be found in the supplementary files. Experimental data, including uncropped western blots, can be found in the Source Data file provided with this paper. Source data are provided with this paper.

## Code availability
Computer code for analysis of simulation trajectories, MD input files, initial and final coordinates are available at https://doi.org/10.5281/ zenodo.10299980.

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

## Acknowledgements

We thank Brandon Liauw and Andrew Hunt for discussions on probing protein-protein interactions, and Katherine Warfel for discussing methods to assess protein expression. This research was supported in part through the computational resources and staff contributions provided for the Quest high performance computing facility at Northwestern University which is jointly supported by the Office of the Provost, the Office for Research, and Northwestern University Information Technology. This work was supported in part by the Searle Funds at The Chicago Community Trust and the National Science Foundation under Grant No. 1844219, 1844336, 2145050. J.A.P. gratefully acknowledges support from the Ryan Fellowship and the International Institute for Nanotechnology at Northwestern University. J.A.P. and T.F.G. were supported by an NSF Graduate Research Fellowship. T.Q.V. was supported by the National Institutes of Health Training Grant (T32GM008449) through Northwestern University's Biotechnology Training Program.

## Author contributions

J.A.P., D.B. and N.P.K. conceived of the idea; J.A.P. and N.P.K. wrote the manuscript; P.L. and D.B. designed proteins; J.S. performed MD simulations, J.A.P., T.Q.V., T.F.G. and V.T.H. designed and performed cell-free protein characterization experiments; all authors analyzed data, discussed results, and commented on the manuscript.

## Competing interests

N.P.K., J.A.P. and J.S. are inventors on a U.S. provisional patent submitted by Northwestern University that covers organizing cell-free expressed membrane proteins in synthetic membranes. D.B. and P.L. are inventors on U.S. patents that cover the computational design of multipass transmembrane proteins and transmembrane pores submitted by the University of Washington. The remaining authors declare no competing interests.
