## [Peer Review File · Nature Communications]

Reviewers' Comments:

Reviewer #1:

Remarks to the Author:

The manuscript by J. Peruzzi et al reports about elegant and systematic approaches that combine MD simulations and synthetic biology to study the effect of hydrophobic mismatch in the insertion, function and organization of transmembrane proteins. As far as I know, this is the first experimental study that provides clear-cut results on this old question.

The authors succeed to integrate their proteins of controlled hydrophobic thickness using a cell-free approach. They study 2 types of proteins, either with 2 spanning helices, or pores. This first part of the paper is already very interesting since it shows the importance of hydrophobic matching for the correct translation of full-length proteins and their membrane insertion using cell-free systems. Data shown in Fig. 2 are also very exciting. They show how hydrophobic mismatch can affect the function of robust proteins such as pores, but also how it is possible to design systems where 2 types of proteins can be preferentially targeted to different liposomes, based on hydrophobic matching. In the last part, the authors study how proteins can be reorganized among different lipid domains, depending in their respective hydrophobic length. Here, I think that the work could be nicely supplemented by experiments showing that protein insertion can drive phase separation in an initially homogeneous mixture, as predicted in their MD simulations. As far as I know, it would also be the first well-controlled realization of the predictions of ref 39, confirmed by simulations of this manuscript. Also, in the simulations related to Fig.3F, it would be interesting to compare the distance between thick and thin proteins in homogeneous membrane, to the distance between proteins of the same type in this membrane.

Altogether, I think that this work will have important insights in synthetic biology, membrane biophysics, but also cell biology. The in vitro assays are well-designed and convincing. I am not an expert in MD simulations, thus cannot provide an opinion on this aspect. I think that the additional experiments and simulations would reinforce the paper, and that it deserves publication in Nature Communications.

Reviewer #2:

Remarks to the Author:

Peruzzi et al present an elegant manuscript that investigates the insertion of several multi-pass transmembrane proteins into synthetic membranes. They show that by tuning the lipid composition to minimize hydrophobic thickness mismatch between the transmembrane domains and the lipid bilayer, they can tune which protein integrate optimally into the lipid vesicles. Furthermore, by incorporating two different proteins with different transmembrane thickness into a phase-separated vesicle which contains two lipid mixtures (and thus membrane thicknesses) they can tune the protein-protein interactions. I think this manuscript is well-presented and well-written, though a bit over-stated and under-analyzed. I do have some concerns as to the methodology and transparency of the data that should be addressed.

Major comments:

The methods need to be more explicit in the actual results and figure legends. For example, instead of saying 23 Å membranes, the composition should also be stated. As it is, in Fig 1E and Supp Fig 2-3 (as just examples), there is no mention of the lipid composition. Given that coarse-grained MD was used, the exact same compositions as was modeled in Fig 1A-C does not appear to be what was used in 1E from the sparse methods section "Vesicle Preparation." A table should be included for each figure that uses the various lipid mixtures stating the compositions explicitly. Furthermore, it is not stated how the thickness of the lipid vesicles is determined. If this is from MD only, this needs to be stated. Better yet would be an actual measurement of the lipid and protein thickness, though I do appreciate the technical hurdles here.

The GUV experiments are not clear. Why is a different lipid and protein used in Fig S3 compared to the rest of the vesicle work? How does the membrane hydrophobic mismatch compare with the 40Å protein used here in a DOPC lipid vesicle? In Fig S11A, this is a poor choice for an image to show that a vesicle is not phase-separated. There is clearly a polarization artifact occurring in your

confocal microscope such that the poles are not being excited to the same extent as the equator. This most frequently occurs in spherical membranes and can be alleviated by including a quarterwave plate in the excitation light path to make the circularly polarized light which is out of plane polarized.

Are the results general (i.e. will the same results hold in Fig 1G if you instead express the 20Å protein in a "thick" membrane?)?

The truncation experiment is not convincing nor easy to follow given the lack of details in the text and figure legend. According to the methods, the authors use an anti-GFP antibody to visualize the synthesized proteins. Why is the FLAG tag included in the construct then? The scheme in 1G needs to include the EGFP if that is indeed what is being visualized. Furthermore, the full blot that is shown does not show any major differences between the full-length and truncation products. Lane 5 shows less truncation products comparable to a similar level of full-length, which is not shown in the graph in Fig 1G. As a minor note, the lipid compositions in the caption of Fig S7 are helpful (and similar to what I was suggesting in my point above) though, I would add in the hypothesized thickness so that the blot can be directly compared to not only Fig 1G, but also the rest of the paper. More experiments need to be done for the authors to conclude that the differences in protein expression are due to translational defects. I find this the least convincing part of this story.

More experimental details need to be shown in the supplement regarding the flow cytometry analysis of the Pacific blue-labeled anti-FLAG bead experiment. All of the gated events for one experiment should be shown in plots in the supplement. It is not clear to me how the calculations of "enrichment in thick membranes" is performed. The statement "suggesting either all proteins have a preference for the thinner membrane" needs to be clarified and addressed as this would go against the thesis of this manuscript which states that proteins prefer to minimize hydrophobic mismatch.

Can the authors comment on why the Cd/Ch value is above 1 for the 50Å protein in Fig 3C? Does this not suggest that the 50Å protein is residing in the Ld (shorter membrane thickness) region? The absolute values for the NBD-Rhodamine look more convincing as to going from phase-separated to mixed. The statement that the FRET data "mirrors" the MD data is overstated, especially if the longer protein really is residing more closely to the dye-labeled 18:1 lipid as it is suggesting with the data.

Is it possible to plot the 50A-50A in the DOPC system here on the same graph (3H) to show that this is indeed a demixing effect? In the 50A-50A in DOPC membranes, the temperature shouldn't have a luciferase-normalized effect, correct? What data is plotted in S17E? Which proteins are analyzed?

Minor comments:

1. At the first mention of "thin, medium, and thick" membranes in the Results, it would be helpful to add in the estimated thicknesses again to aid those readers that might skip over the introduction.
2. The cartoon in 2G is the opposite of what is shown in 2H. Do proteins functionally insert into thinner membranes or thicker membranes? Maybe I'm confused considering the calculations for the y-axis of 2F and 2H are not given anywhere.
3. The manufacturer of all materials needs to be listed (especially the antibodies used as they are highly variable).
4. To fully understand Fig S13, the exact nature of the lipids and FRET pair needs to be stated. NBD-labeled lipids are never mentioned in the Methods. Are these head group-labeled or tail-labeled? Is it PC or PE-rhodamine?
5. Again, the lipid compositions of the modeled systems need to be listed clearly in the Results

and figure legends. This statement is unacceptable: "We simulated membrane interactions with thin (20 Å) and thick (50 Å) proteins using coarse-grained MD simulations of lipid composition comparable to the experimental system." What is comparable? How is the "mol Fraction DyPC in contact with protein" calculated in Fig 3A? Perhaps the authors can instead calculate number of DyPC or DPPC in contact with protein.

6. Why don't the values for the domains 47C in Fig S15 add up to 1 like they should? Something seems to be off in this plot. Perhaps a finer binning than 5nm would help to make the point more clear. What do these distances represent in terms of the size of the box modeled. Is 5nm "close"? If so, how many lipids away?

7. What is the concentration of BSA used in coating the glass coverslips for the GUVs?

Reviewer #3:

Remarks to the Author:

The manuscript by Peruzzi et al demonstrates how hydrophobic mismatch can be exploited to tune protein integration and organization in model membranes using de novo protein design, molecular dynamics simulations, and cell-free systems. The hydrophobic mismatch is among the effective membrane-mediated interactions that are also believed to be involved in cellular membrane organization in living cells. Membrane lateral organization and mechanisms that govern their organization are interesting and important topics. Combining these techniques is a robust scheme to explore such mechanisms. However, prior to making any decision regarding the acceptance of the article, I believe there are several important points that should be addressed. Therefore, I will reconsider my decision once I have received answers to the following comments.

My main problem is the presentation of the simulation results.

Molecular dynamics simulations have been performed by the authors, but there is no snapshot of the simulated systems (except for Figure SI-12, which is also unclear). As an example, the cartoon figures in figures 1 and 3 (in particular figure 1A) should be replaced by snapshots of the simulated system. A table should be provided (either in the SI or in the main text) indicating all simulated systems and the amount of time that has been simulated.

More comments:

As it is now, it seems that the simulations are only 6microseconds long. Does 6 microseconds is enough to observe phase separation in ternary mixture systems? Because Figure SI-12 does not show a phase-separated membrane. Moreover, in Figure SI-12 both DPPC and CHOL are shown with the same color which makes it even more unclear.

Fig 4A has been mentioned in the below sentence but I could not find it.

"For simulations shown in Fig. 4A, membranes were composed of 138 DPPC, 92 DYPC and 99 cholesterol molecules per leaflet"

In this study, the authors investigated the effects of temperature change using the Martini model. The authors should provide some articles that demonstrate the Martini model is temperature transferable or, if not, explain why this won't be relevant to their study.

The authors write "However, it has also been hypothesized that more passive lipid-protein interactions can drive inter- and intramembrane protein organization" and cites several articles that only a few are relevant and relevant existing literature are missing. For example, see several works by Johannes group:

Johannes et al, Clustering on membranes: fluctuations and more; Trends in cell biology 28 405-415 2018

Pezeshkian et al, Mechanism of Shiga toxin clustering on membranes; ACS nano 11 314-324 (2017)

Arumugam et al, Ceramide structure dictates glycosphingolipid nanodomain assembly and

function, Nature communications 12, 1-12 (2021)

REVIEWER COMMENTS

Reviewer #1 (Remarks to the Author):

The manuscript by J. Peruzzi et al reports about elegant and systematic approaches that combine MD simulations and synthetic biology to study the effect of hydrophobic mismatch in the insertion, function and organization of transmembrane proteins. As far as I know, this is the first experimental study that provides clear-cut results on this old question.

The authors succeed to integrate their proteins of controlled hydrophobic thickness using a cell-free approach. They study 2 types of proteins, either with 2 spanning helices, or pores. This first part of the paper is already very interesting since it shows the importance of hydrophobic matching for the correct translation of full-length proteins and their membrane insertion using cell-free systems. Data shown in Fig. 2 are also very exciting. They show how hydrophobic mismatch can affect the function of robust proteins such as pores, but also how it is possible to design systems where 2 types of proteins can be preferentially targeted to different liposomes, based on hydrophobic matching. In the last part, the authors study how proteins can be reorganized among different lipid domains, depending in their respective hydrophobic length. Here, I think that the work could be nicely supplemented by experiments showing that protein insertion can drive phase separation in an initially homogeneous mixture, as predicted in their MD simulations. As far as I know, it would also be the first well-controlled realization of the predictions of ref 39, confirmed by simulations of this manuscript.

We thank the reviewer for this suggestion. We agree that demonstrating that phase separation can be driven by protein insertion experimentally would be a powerful addition to this study. To explore this behavior experimentally we used lipid-lipid FRET. We prepared ternary lipid membranes (42.5 mol% 14:1 PC/27.5 mol% DPPC/30 mol% Chol; 14:1 PC and DPPC have thicknesses of 23.4 and 34.4 Å respectively) labeled with 0.1 mol% 18:1 PE Rhodamine and 16:0 PE NBD. We expressed each protein into these lipid mixtures and then measured lipid-lipid FRET at 20°C and 50°C, below and above the phase transition temperature respectively. Based on our simulations, we hypothesized that the lipid dyes, which should segregate to different lipid phases, would report changes in protein-induced membrane phase separation. Mainly, the dyes should remain farther apart, with a corresponding lower C_D/C_H value, when proteins with increased hydrophobic mismatch are integrated into vesicles relative to vesicles with no protein or proteins with shorter transmembrane spans. Interestingly, we observed this behavior – the membranes with the 50 Å protein had lower C_D/C_H values at 20 and 50°C compared to samples with no protein or the 20 Å thick protein, suggesting that domains formed to a greater extent when the 50 Å protein was inserted into membranes, and leading to greater distances between the rhodamine and NBD dyes. However, although the mean values reflect this trend, they are not significantly different.

Revision Figure 1. C_D/C_H for samples with no protein, 20 Å, and 50 Å proteins at 20°C and 50°C. $n=3$. Proteins were expressed into ternary lipid membranes (42.5 mol% 14:1 PC/27.5 mol% DPPC/30 mol% Chol) labeled with 0.1 mol% 18:1 PE Rhodamine and 16:0 PE NBD. C_D/C_H values for vesicles with no protein, the 20 Å, and the 50 Å protein were measured at 20 °C and 50°C.

This trend can be further observed by plotting $F_{donor}/F_{acceptor}$ for the ternary membranes. This value should increase as the lipid dyes move farther apart on average within a vesicle membrane. Here, the trace for the 50 Å membrane curves upward, diverging from the water and 20 Å protein samples as temperature is increased above the phase transition temperature of the phase segregated lipid vesicle ($T_m=41^\circ\text{C}$ for DPPC). This shift in FRET signal suggests that the lipid probes remain farther away from one another at elevated temperatures. This increased distance could be due to the fact that domains are forming around the 50 Å protein. The enhanced differences in FRET signals between the 50 Å protein containing membranes and other membranes at higher temperatures may be due to the 50 Å protein supporting the local presence DPPC and 16:0 NBD rich lipid domains, thus causing $F_{donor}/F_{acceptor}$ to deviate from the no protein and 20 Å protein containing membranes.

Revision Figure 2. $F_{donor}/F_{acceptor}$ values for 42.5 mol% DyPC/27.5 mol% DPPC/30 mol% Chol membranes with no protein (water), 20 Å protein, and 50 Å protein. Membranes were labeled with 0.1 mol% 18:1 PE Rhodamine (acceptor) and 0.1 mol% 16:0 PE NBD (donor) and read over temperature.

From these experiments, we believe that the phenomena which was predicted via simulation is likely occurring, however because the C_D/C_H values are not significantly different we believe that more work must be done to demonstrate this phenomenon experimentally and are actively researching other experimental techniques (ex. Small-angle X-ray scattering) that might help us assess nanoscale domain formation around proteins in future work.

Also, in the simulations related to Fig.3F, it would be interesting to compare the distance between thick and thin proteins in homogeneous membrane, to the distance between proteins of the same type in this membrane.

We thank the reviewer for the suggestion which makes a great addition to the study. We have run the requested simulation and additionally, we have performed additional replicates for the 20 Å – 50 Å protein pair (see updated Figure below). Consistently, we see that the tertiary lipid mixture inhibits close contact for proteins of different thicknesses and favors association of equal length proteins. (Methods:) Simulations of homotypic and heterotypic pairs of the 50 Å and 20 Å proteins were run in single component (DOPC, no domains) or tertiary membranes (DYPC/DPPC/Chol, domains). The changes in protein association were quantified by calculating the pseudo free energy difference between the membranes assembled from ternary and single component membranes as $-\log(N_{\text{bound,ternary}}/N_{\text{bound,single}})$, where N is the number of states which we considered ‘bound’. Proteins were considered ‘bound’ if their center-to-center distance was below 3 nm. Standard errors were calculated from simulation replicates. Using this metric, a negative pseudo free energy difference suggests that proteins interact more in ternary membranes compared to the homogenous membrane. We observe negative values for $\log(N_{\text{bound,ternary}}/N_{\text{bound,single}})$ for pairs of proteins with the same hydrophobic thickness (20 Å – 20 Å and 50 Å- 50 Å) and a positive value of $\log(N_{\text{bound,ternary}}/N_{\text{bound,single}})$ when proteins possess different hydrophobic thicknesses (20 Å – 50 Å). These results demonstrate that proteins of different transmembrane domain lengths are farther apart in the ternary lipid mixture relative to the homogenous membrane but proteins of equal transmembrane domain length are closer together in ternary membranes relative to homogenous membranes. This is fully consistent with our results and a previous computational study considering gel lipid membranes with fluid domains.¹

Supplementary Figure 15. MD simulations predict protein-protein in synthetic membranes. (A) Simulations of homotypic and heterotypic pairs of the 50 Å and 20 Å proteins were run in single component (DOPC, no domains) or tertiary membranes (42.5 mol% DyPC/27.5 mol% DPPC/30 mol% Cholesterol, domains). The changes in protein association were quantified by

calculating the pseudo free energy difference between the protein-protein tertiary and single component membranes as $-\log(N_{\text{bound,ternary}}/N_{\text{bound,single}})$, where N is the number of states which we considered bound. Proteins were considered bound if their center-to-center distance was below 3 nm. Standard errors were calculated from simulation replicates. Negative values represent enhanced protein-protein contact formation in ternary membranes compared to single component membranes. These data demonstrate that proteins of different transmembrane domain lengths are farther apart in the ternary lipid mixture relative to the homogenous membrane while proteins of equal transmembrane domain length are closer together in ternary membranes relative to homogenous membranes.¹ **(B)** Protein-protein distance of 20 and 50 Å proteins in DOPC and domain forming (42.5 mol% DyPC/27.5 mol% DPPC/30 mol% Cholesterol) lipid mixtures at 25°C and 47°C. At 25 °C, protein-protein distance between the 20 and 50 Å hairpin is on average smaller in homogenous, single component DOPC membranes compared to membranes composed of 42.5 mol% DyPC/27.5 mol% DPPC/30 mol% Cholesterol. At 47°C, protein-protein distance decreases in membranes composed of 42.5 mol% DyPC/27.5 mol% DPPC/30 mol% Cholesterol due to increased lipid mixing. Histograms were generated from 3 independent simulations. Bin size is 3 nm, the approximate protein center to center distance when bound.

Altogether, I think that this work will have important insights in synthetic biology, membrane biophysics, but also cell biology. The in vitro assays are well-designed and convincing. I am not an expert in MD simulations, thus cannot provide an opinion on this aspect. I think that the additional experiments and simulations would reinforce the paper, and that it deserves publication in Nature Communications.

Thank you for your thoughtful comments and insight!

Reviewer #2 (Remarks to the Author):

Peruzzi et al present an elegant manuscript that investigates the insertion of several multi-pass transmembrane proteins into synthetic membranes. They show that by tuning the lipid composition to minimize hydrophobic thickness mismatch between the transmembrane domains and the lipid bilayer, they can tune which protein integrate optimally into the lipid vesicles. Furthermore, by incorporating two different proteins with different transmembrane thickness into a phase-separated vesicle which contains two lipid mixtures (and thus membrane thicknesses) they can tune the protein-protein interactions. I think this manuscript is well-presented and well-written, though a bit over-stated and under-analyzed. I do have some concerns as to the methodology and transparency of the data that should be addressed.

Major comments:

The methods need to be more explicit in the actual results and figure legends. For example, instead of saying 23 Å membranes, the composition should also be stated. As it is, in Fig 1E and Supp Fig 2-3 (as just examples), there is no mention of the lipid composition. Given that coarse-grained MD was used, the exact same compositions as was modeled in Fig 1A-C does not appear to be what was used in 1E from the sparse methods section "Vesicle Preparation." A table should be included for each figure that uses the various lipid mixtures stating the compositions explicitly

We appreciate the reviewer's advice and have added more description in the results and figure captions to more clearly specify the membrane compositions used. Further we have added two tables- one detailing the phase transition temperature and thickness of lipids used in this study, and a second table summarizing membrane compositions in the supplement.

Supplementary Table 1. Physical features of lipids used in this study. Reported transition temperatures (T_m) are listed by Avanti Polar Lipids, from which the lipids were purchased.

Lipid	T _m (°C)	Thickness (Å)
14:1 PC	n/a	23.4 ²
18:1 PC (DOPC)	-17	29.3 ²
22:1 PC	13	37.2 ²
16:0 PC (DPPC)	41	34.4 ³

Supplementary Table 3. Table of lipid compositions used.

Figure	Panel	Composition
Fig. 1	b	DYPC (di-C12:1-C14:1 PC), DOPC (18:1 PC), DGPC (di-C20:1-C22:1) (simulation)
	c	DYPC, DOPC, DGPC (simulation)
	e	Water, 14:1 PC, DOPC, 22:1 PC
	f	Comparison of simulated membrane compression in DYPC, DOPC, DGPC to GFP fluorescence in 14:1 PC, DOPC, 22:1 PC membranes
	g	Water, 14:1 PC, DOPC, 22:1 PC
Fig. 2	c	14:1 PC
	d	22:1 PC
	f	14:1 PC and 22:1 PC
	h	14:1 PC and 22:1 PC
Fig. 3	a	42.5 mol% DYPC/27.5 mol% DPPC (16:0 PC)/30 mol% Cholesterol
	c	42.5 mol% 14:1 PC/27.5 mol% DPPC/30 mol% Cholesterol
	d	42.5 mol% 14:1 PC/27.5 mol% DPPC/30 mol% Cholesterol
	f	42.5 mol% 14:1 PC/27.5 mol% DPPC/30 mol% Cholesterol and DOPC
	h	42.5 mol% 14:1 PC/27.5 mol% DPPC/30 mol% Cholesterol and DOPC
Supplementary Fig. 1	b-d	DYPC, DOPC, DGPC (simulation)
Supplementary Fig. 2	a-h	Water, 14:1 PC, DOPC, 22:1 PC
Supplementary Fig. 3		99.9 mol% DOPC, 0.1 mol% 18:1 PE Cy5.5

Supplementary Fig. 4		Water, 14:1 PC, DOPC, 22:1 PC
Supplementary Fig. 5		Water, 14:1 PC, DOPC, 22:1 PC
Supplementary Fig. 6		Water, 14:1 PC, DOPC, 22:1 PC
Supplementary Fig. 7		Water, 14:1 PC, DOPC, 22:1 PC
Supplementary Fig. 8		DOPC
Supplementary Figure 9		99.9 mol% 14:1 PC, 0.1 mol% 18:1 PE Cy5.5; 99.9 mol% 22:1 PC, 0.1 mol% 18:1 PE Rhodamine
Supplementary Figure 10	a, b	99.9 mol% 14:1 PC, 0.1 mol% 18:1 PE Cy5.5; 99.9 mol% 22:1 PC, 0.1 mol% 18:1 PE Rhodamine
Supplementary Figure 11	a	42.5 mol% 14:1 PC/27.5 mol% DPPC/30 mol% Chol + 0.1 mol% 18:1 PE Rhodamine
	b	40 mol% 14:1 PC/40 mol% DPPC/20 mol% Chol + 0.1 mol% 18:1 PE Rhodamine
Supplementary Figure 12		42.5 mol% DYPC/27.5 mol% DPPC/30 mol% Cholesterol
Supplementary Figure 13	b	DOPC and 42.5 mol% 14:1 PC/27.5 mol% DPPC/30 mol% Cholesterol with 0.1 mol% 18:1 PE Rhodamine and either 0.1 mol% 18:1 PE or 16:0 PE NBD
Supplementary Figure 14	a	99.9 mol% DOPC, 0.1 mol% 18:1 PE Rhodamine
	b-c	42.5 mol% 14:1 PC/27.5 mol% DPPC/30 mol% Cholesterol with 0.1 mol% 18:1 PE Rhodamine
Supplementary Figure 15		DOPC and 42.5 mol% DYPC/27.5 mol% DPPC/30 mol% Cholesterol (Domains) (Simulation)
Supplementary Figure 16	b	DOPC and 42.5 mol% 14:1 PC/27.5 mol% DPPC/30 mol% Cholesterol (Domains)
Supplementary Figure 17	a-e	DOPC and 42.5 mol% 14:1 PC/27.5 mol% DPPC/30 mol% Cholesterol (Domains)

Also, you are correct – the exact compositions were not matched between experiment and simulation because for simulations, we relied on available Martini lipids. In simulations, DyPC (di-C12:1-C14:1 PC) was

used in place of 14: PC (di-C14:1-C14:1) and DGPC (di-C20:1-C22:1) was used instead of 22:1 PC (di-C22:1-C22:1) due to their availability in the Martini force field. We believe that these differences are small and that the simulations are still reflective of the experimental system. The choice of lipids for simulations versus experiments has been more explicitly stated in the methods:

“In simulations, DyPC (di-C12:1-C14:1 PC) was used in place of 14: PC (di-C14:1-C14:1) and DGPC (di-C20:1-C22:1) was used instead of 22:1 PC (di-C22:1-C22:1) due to the availability of Martini lipids. We believe that these differences are small and that the simulations are still reflective of the experimental system.”

Furthermore, it is not stated how the thickness of the lipid vesicles is determined. If this is from MD only, this needs to be stated. Better yet would be an actual measurement of the lipid and protein thickness, though I do appreciate the technical hurdles here.

For experimental studies, membrane thicknesses were determined by Heberle et al. (2020) using small-angle x-ray scattering.² For calculating membrane compression, membrane thicknesses were determined by simulation. We have added statements to clarify this in the methods:

“Reported membrane thicknesses for 14:1 PC, DOPC, and 22:1 PC were determined by Heberle et al. using small-angle x-ray scattering².”

The GUV experiments are not clear. Why is a different lipid and protein used in Fig S3 compared to the rest of the vesicle work? How does the membrane hydrophobic mismatch compare with the 40 Å protein used here in a DOPC lipid vesicle? In Fig S11A, this is a poor choice for an image to show that a vesicle is not phase-separated. There is clearly a polarization artifact occurring in your confocal microscope such that the poles are not being excited to the same extent as the equator. This most frequently occurs in spherical membranes and can be alleviated by including a quarterwave plate in the excitation light path to make the circularly polarized light which is out of plane polarized.

The lipid and protein compositions in Figure S3 are the same as that used with the 40 Å thick protein and 29 Å thick membrane shown in Fig. 1E. The only difference is that in Figure S3, we chose to use Cy5.5 PE to stain the DOPC membrane, as its excitation and emission are farther away from mEGFP (fused to the 40 Å protein) compared to Rhodmaine and NBD labeled lipids used throughout the work. We have added the following statement to the figure caption to clarify this:

“This composition corresponds to the 29 Å thick membrane and 40 Å thick protein in Fig. 1E.”

Additionally, we have updated Figure S11 to a better image and have added a quarterwave plate in the excitation light path of our microscope (Thanks for the suggestion!).

Supplementary Figure 11. Fluorescent microscopy of giant unilamellar vesicles demonstrates that lipid mixtures do not form microdomains. (A) Vesicles composed of 42.5 mol% 14:1 PC/27.5 mol% DPPC/30 mol% Chol (composition used in this study) and (B) 40 mol% 14:1 PC/40 mol% DPPC/20 mol% Chol. Both membranes in (A) and (B) were labeled with 0.1 mol% 18:1 PC Rhodamine, which localized to the lipid disordered phase. Exclusion of dye in a region of the membrane, as seen in (B), indicates the presence of microdomain formation, a property often seen in previous studies. By increasing the cholesterol content and decreasing the amount of DPPC, membranes that do not exhibit microdomain formation (A) are formed. Samples with higher cholesterol content used to make vesicles in panel A do not exhibit microdomain formation. Scale bars are 10 μm .

Are the results general (i.e. will the same results hold in Fig 1G if you instead express the 20Å protein in a “thick” membrane?)?

We thank the reviewer for raising this question. We have repeated this experiment by performing western blots on cell-free reactions containing vesicles and on vesicles purified away from the reactions and non-associated protein following expression of the the 20 Å, 28 Å, 32 Å, and 50 Å proteins in 14:1 PC, 18:1 PC, and 22:1 PC membranes. Overall, we found that while the truncation pattern observed for the 50 Å protein was reproducible, the associated trend of increased truncation products with increased hydrophobic mismatch does not hold true for the other proteins tested. If true, one would expect that the ratio of full-length to truncated peptides would be highest when each protein was expressed into the hydrophobically matched membrane. This was not observed across the panel of new proteins tested, thus we have altered the language to pose this as a hypothesis and have moved all related data to the supplement. Based on our experiments, proteins do express and insert better into hydrophobically matched membranes. Thus, increased hydrophobic mismatch must impose some sort of inhibitory feedback on the cell-free system. We hypothesize that hydrophobic mismatch decreases the ability of a protein to stably insert into a membrane. These uninserted proteins likely aggregate, and reduce the cell-free system’s capacity to produce protein. Moving forward, we plan to investigate this possibility further with assays more sensitive than the semi-quantitative western blots we used.

Revision Figure 3. Analysis of the truncation products formed by the 20 Å protein in cell-free reactions (top) and associated with vesicles (bottom). Unpurified samples reflect western blots from the reaction mixture which contain both vesicles and cell-free reaction components. Purified samples reflect samples after purification of vesicles away from non-integrated proteins and the reaction components.

Revision Figure 4. Analysis of the truncation products formed by the 28 Å protein in cell-free reactions (top) and associated with vesicles (bottom).

32 Å Protein - Unpurified

32 Å Protein - Purified

Revision Figure 5. Analysis of the truncation products formed by the 32 Å protein in cell-free reactions (top) and associated with vesicles (bottom).

New Supplementary Figure:

Supplementary Figure 7. Analysis of truncation products via western blot. (A) An N-terminal flag tag was added to the 50 Å hairpin protein enabling the detection of all protein products. (B) The construct was expressed in the presence of water, 14:1 PC (23 Å), 18:1 PC (29 Å), and 22:1 PC (37 Å) and protein formation was assessed via western blot. (C) The proportion of full length 50 Å protein relative to incomplete proteins, increases as hydrophobic mismatch is minimized. Protein expression and truncation products were assessed by performing a Western Blot against a N-terminal FLAG tag. (D, E) Western blot and quantification of full-length and truncated protein products associated with purified membranes. To observe full-length protein and truncation products associated with vesicles, 1 mol% Biotinyl Cap PE was included in each membrane. 3.5 µL of cell-free reaction mixture was mixed with 0.1 mg of Pierce Streptavidin beads to capture vesicles. The bead mixture was then resuspended and loaded into the gel. The intensity of the full-length product and truncation products, as noted by the labeled arrows, was measured using ImageJ and the ratio of full-length to truncation product intensity. (F) Schematic illustrating how hydrophobic mismatch may lead to an increase truncated proteins products and subsequent decreased in protein expression.

The truncation experiment is not convincing nor easy to follow given the lack of details in the text and figure legend. According to the methods, the authors use an anti-GFP antibody to visualize the synthesized proteins. Why is the FLAG tag included in the construct then? The scheme in 1G needs to include the EGFP if that is indeed what is being visualized. Furthermore, the full blot that is shown does not show any major differences between the full-length and truncation products. Lane 5 shows less truncation products comparable to a similar level of full-length, which is not shown in the graph in Fig 1G. As a minor note, the lipid compositions in the caption of Fig S7 are helpful (and similar to what I was suggesting in my point above) though, I would add in the hypothesized thickness so that the blot can be directly compared to not only Fig 1G, but also the rest of the paper. More experiments need to be done for the authors to conclude

that the differences in protein expression are due to translational defects. I find this the least convincing part of this story.

For truncation experiments, we added a FLAG tag to the N-terminus of the protein so that we could observe truncation products, identified by detecting proteins with the N-terminus Flag tag. Full length products are identified by detecting proteins with the C-terminus GFP, but products that terminated early during translation will not have the complete GFP fusion. Therefore, for truncation experiments, we did not probe expression with an anti-EGFP antibody so the scheme in Fig 1G is correct. This approach was stated in the main text and we have added an additional statement to clarify our use of the N terminal tag for truncation products:

“To characterize how translation is affected by membrane-protein hydrophobic mismatch, we added a N-terminal FLAG tag to the 50 Å protein, allowing us to monitor the formation of truncated protein products in addition to full-length protein products identified through the C-terminus EGFP. By probing the FLAG-tag located on the N-terminus of the protein produced in PURExpress, both full-length and partial translated product can be observed via Western Blot.”

The PURExpress system used in this study is comprised of purified proteins required for protein transcription and translation and thus does not contain proteases or cellular machinery that would degrade/get rid of incomplete protein products. By probing the FLAG-tag located on the N-terminus of the protein produced in PURExpress, both full-length and partial translated product can be observed via Western Blot. We agree that this is not the best method to demonstrate protein truncation. We initially tried to detect incomplete translation by incorporating both radiolabeled and fluorescent amino acids into the proteins. Unfortunately, the autoradiogram and fluorescent gels were not sensitive enough to adequately observe truncation products so we turned to luminescence based Western Blots.

In the meantime, to further investigate the potential of translational defects to contribute to the preferred cell free expression of proteins into hydrophobically matched membranes, we developed a quantitative model to describe the role of membranes in the cell-free expression of membrane proteins using another model membrane protein (currently in revision as a separate manuscript). In this study, we further quantified incomplete translation of membrane proteins in cell-free systems⁴. We show a correlation between incomplete translation (measured by monitoring truncation products) and reduced protein expression for several model membrane proteins, compared to soluble proteins in cell-free systems. We developed a kinetic model to describe this experimentally observed phenomena that the reduced expression of full-length membrane protein is due to aggregation of protein, and believe a similar kinetic phenomenon is at play here where hydrophobic mismatch impacts expression of membrane proteins. It is however possible that additional mechanisms contribute to the reduction of full length protein with hydrophobic mismatch. For example, Choi et al. estimate a membrane insertion energy of about 10-20 kBT per harpin, comparable to membrane deformation energies⁵. Thus, even full-length protein might become destabilized with increasing hydrophobic mismatch, leading to reduction of folded and membrane inserted protein. We have added this discussion to the main text:

We then set out to characterize how translation is affected by membrane-protein hydrophobic mismatch. In separate work, we have demonstrated that incomplete translation is higher and overall expression of several model membrane proteins are lower compared to soluble proteins in cell-free systems. We attribute this to aggregation of protein through the development of a kinetic model, and believe a similar kinetic phenomenon affects protein expression as a function of hydrophobic mismatch⁴. To monitor this, we added an N-terminal FLAG tag to the 50 Å protein, allowing us to observe the formation of truncated protein products in addition to full-length protein products identified through the C-terminus mEGFP. Analyzing the total expression of 50 Å protein in cell free reactions and protein associated with vesicles by western blot, we observed an increase in incomplete protein products relative to full-length protein as a function of

hydrophobic mismatch (Supplementary Figure 7). The higher proportion of truncated protein products in the more hydrophobically mismatched system suggests that translation is affected by hydrophobic mismatch. We hypothesize this effect arises from the energy cost of deforming membranes to accommodate differences in hydrophobic mismatch, which reduces the probability of protein co-translational insertion and proper folding. Misfolded proteins likely capture nascent proteins from the ribosome, thus reducing the rate of protein insertion and folding, and increasing the frequency of incomplete translation⁴.

More experimental details need to be shown in the supplement regarding the flow cytometry analysis of the Pacific blue-labeled anti-FLAG bead experiment. All of the gated events for one experiment should be shown in plots in the supplement. It is not clear to me how the calculations of “enrichment in thick membranes” is performed. The statement “suggesting either all proteins have a preference for the thinner membrane” needs to be clarified and addressed as this would go against the thesis of this manuscript which states that protein prefer to minimize hydrophobic mismatch.

We thank the reviewer for this comment. We have added more data to the SI Figure (below) and added more text in the results and methods section. “Enrichment in thick membrane” here is defined as the ratio of Rhodamine mean fluorescence intensity (22:1 PC) to Cy5.5 mean fluorescence intensity (14:1 PC). One would hypothesize that for thinner proteins (20 and 24 Å) the value for enrichment in the thick membrane would be below 1 and for thicker proteins (40 and 50 Å) the value for enrichment in the thick membrane would be above 1. Interestingly, this ratio was below 1 for all four proteins tested but increased with protein thickness. This suggests that if you increase protein hydrophobic thickness, you can bias protein insertion towards thicker membranes. We are not certain however why all of the values are below one. All the values could be below 1 due to all proteins preferring the thinner 14:1 PC membranes. This would contradict data in Figure 1 and in Figure 3H. However, we believe the expression of the proteins (shown in Figure 1 and SI Figure 5) are the most conclusive evidence that proteins are folding and expressing better in hydrophobically matched membranes. Because the experiment with anti-FLAG beads relies on fluorescence signals from vesicle membranes, differential partitioning or dye transfer due to a preference of the membrane dye into the thinner versus thicker membranes may lead to the observed result of more “thin” vesicles being localized to the beads than “thick” vesicles, even if less membrane protein-antibody binding interactions are occurring with proteins in the thin vesicles. Both lipid dyes are conjugated to 18:1 PE and could prefer 22:1 PC membranes compared to 14:1 PC membranes. If this is true, the dye labeling 14:1 PC could transfer to the thicker membranes and cause the overestimation in proteins inserting into thinner membranes. Alternatively, there could vesicle fusion occurring during the cell-free reaction membranes since protein insertion during cotranslational integration should introduce points of membrane instability and membrane defects. We are not able to parse out the exact reason for why it appears lower values of thick vesicles bind the bead (mediated through integrated protein binding to bead-bound antibodies), but would like to emphasize that regardless of the starting values, we still observe a shift in preference with increasing hydrophobic thickness. We furthermore are able to build upon this experiment and observe a shift in pore insertion and material transfer that improves with hydrophobically matched proteins and membranes in Figure 3H.

Supplementary Figure 9. Gating strategy for bead-based protein sorting assays (Figure 3C and 3D). Magnetic protein A/G beads were bound with Pacific Blue conjugated anti-FLAG antibody. Beads were first gated on sizes (1) to analyze larger beads and reduce analysis of unbound vesicles, then gated by the presence of antibody (2). Beads were then gated to analyze single beads (3) and then analyzed for Rhodamine and Cy5.5 fluorescence. First, vesicle controls were analyzed to confirm that the assay was able to detect differences in protein insertion into different vesicle populations. To do this, the 40 Å protein was expressed into DOPC vesicles labeled with either 0.1 mol% 18:1 PE Rhodamine or 0.1 mol% 18:1 PE Cy5.5. By expressing the protein into both samples composed of the same lipid, only differing in dye, we could assume that the protein insertion into both was equivalent. Each vesicle set was then mixed in defined ratios (100% Cy5.5, 0% Rhodamine; 0% Cy5.5, 100% Rhodamine; 75% Cy5.5, 25% Rhodamine; 50% Cy5.5, 50% Rhodamine; 25% Cy5.5, 75% Rhodamine), bound to beads, and then analyzed via flow cytometry. A shift in Rhodamine and Cy5.5 fluorescence which corresponded to the initial ratio was observed. This experiment was then repeated with rhodamine labeled 22:1 PC vesicles and Cy5.5 labeled 14:1 PC vesicles. 22:1 PC and 14:1 PC vesicles were mixed 1:1 and each protein was expressed with this mixture of vesicles. Samples were then bound to beads and analyzed via flow cytometry (5) and enrichment in the thicker, 22:1 PC membranes was calculated (6).

Additional text:

We first validated this method by separately expressing the 40 Å hairpin protein into DOPC vesicles either labeled with 0.1 mol% 18:1 PE Rhodamine or 18:1 PE Cy5.5 dyes. We then mixed different ratios of these protein-integrated vesicles together, bound them to anti-FLAG beads, and analyzed their fluorescence via flow

cytometry. We confirmed that shifts in fluorescence reflected the defined ratio of vesicles added to the beads (Supplementary Figure 9). We then expressed either the 20, 24, 40, or 50 Å thick protein in the presence of a 1:1 mixture of thick and thin membranes (22:1 PC, 37 Å and 23 Å, 14:1 PC respectively). We added anti-FLAG beads to the vesicle mixture after protein expression and integration and measured the ratio of fluorescence from membrane dye in the thick membranes to membrane dye in the thin membranes that was colocalized to the beads.

We found that as hydrophobic thickness of a protein increased, this ratio increased. Interestingly, this ratio was always below one for all proteins tested, suggesting all proteins have a preference for the thinner membrane, which would contradict the protein expression data we previously observed (Fig. 1). Alternatively, the FLAG tag on vesicle-integrated protein might be more accessible when in the thinner membrane, or there may be a propensity for the dye in the thinner membrane (18:1 PE conjugated Cy5.5) to transfer to the thicker membrane, or vesicle instabilities may occur during the cell-free reaction leading to vesicle fusion.

Can the authors comment on why the Cd/Ch value is above 1 for the 50Å protein in Fig 3C? Does this not suggest that the 50Å protein is residing in the Ld (shorter membrane thickness) region? The absolute values for the NBD-Rhodamine look more convincing as to going from phase-separated to mixed. The statement that the FRET data “mirrors” the MD data is overstated, especially if the longer protein really is residing more closely to the dye-labeled 18:1 lipid as it is suggesting with the data.

A Cd/Ch value above 1 does suggest that the 50 Å protein interacts with the rhodamine labeled lipid in the ternary lipid mixture more than the homogeneous lipid mixture. We would like to emphasize that membranes are not phase separated in the traditional sense with a stable liquid ordered and disordered phase. We do not observe micro domains in GUVs and in simulations, lipid organization and sorting is only observed when proteins are present in the membrane. As temperature is increased, Cd/Ch for the 50 Å protein increases suggesting that at elevated temperatures its interaction with 18:1 PE Rhodamine increases. We have reworded this section to more accurately reflect the data:

“We measured C_D/C_H values of the 20 and 50 Å protein over a range of temperatures. At room temperature, the 20 Å protein had a higher C_D/C_H , indicating that it resides in the dye containing, thinner and more unsaturated 14:1 PC rich phase. Upon increasing temperature to dissolve domains, we observed that the average distance between the unsaturated lipid dye and thinner protein increased. Conversely, the 50 Å protein had lower C_D/C_H values at lower temperatures that increased with increasing temperature. This result suggests that the 50 Å protein interacts more with the rhodamine-labeled lipid than the NBD-labeled lipid when present in a tertiary composition membrane than when present in the homogenous one. The membranes we generated for experimental studies are not phase separated in the traditional sense with a stable liquid ordered and disordered phase, especially because we conduct these studies close to the melting transition temperature of the tertiary lipid composition. We do not observe micro domains in GUVs and in simulations in the absence of proteins. In contrast, phase separation or an increase in lipid organization is only observed when proteins are present in the membrane. Nevertheless, the increase in C_D/C_H values as temperature is increased suggests the larger 50 Å protein moves from a state where its interactions with the dye poor, thicker and more saturated DPPC rich lipid phase decrease and the protein interacts more with the dye-rich, thinner and more unsaturated 14:1 PC lipids (Fig. 3C). The FRET data obtained as a function of temperature for the 20 Å protein mirrors our simulation data in Fig. 3A.^{6,7} However, domains around the 50 Å protein in the simulation appear to be more stable than in the synthetic membrane experiments, likely due to different parameters between the MARTINI model and experimental conditions.”

Is it possible to plot the 50A-50A in the DOPC system here on the same graph (3H) to show that this is indeed a demixing effect? In the 50A-50A in DOPC membranes, the temperature shouldn't have a luciferase-normalized effect, correct? What data is plotted in S17E? Which proteins are analyzed?

The data plotted in 3H includes the traces for the 50 Å-50 Å proteins in DOPC membranes. 3H plots relative NanoBit assembly, defined as:

To characterize protein-protein interactions with increasing temperature, the luminescence of samples was then recorded at varying temperatures from room temperature to 45 °C. Relative NanoBit assembly was then calculated as:

$$\text{Relative NanoBit Assembly} = \frac{\text{Lum}_{20\text{\AA}-50\text{\AA}}}{0.5 * (\text{Lum}_{20\text{\AA}-20\text{\AA}} + \text{Lum}_{50\text{\AA}-50\text{\AA}})}$$

Where $\text{Lum}_{20\text{\AA}-50\text{\AA}}$ is the luminescence of samples with 20 Å and 50 Å hairpin proteins, $\text{Lum}_{20\text{\AA}-20\text{\AA}}$ is the luminescence of samples with 20 Å and 20 Å hairpin proteins, and $\text{Lum}_{50\text{\AA}-50\text{\AA}}$ is the luminescence of samples with 50 Å and 50 Å hairpin proteins. Luminescence values were then normalized to the luminescence value at room temperature.

All luciferase data is affected by temperature as luciferase activity decreases above room temperature. Raw data for split luciferase fused to membranes is plotted in Supplementary Figure 17A and soluble luciferase activity over temperature is plotted in Fig. S17B. We have updated Figure S17E to compare luminescence of the 20 Å- 50 Å, 20 Å-20 Å, and 50 Å-50 Å proteins in the ternary lipid membrane (42.5% 14:1 PC/27.5% DPPC/30% Chol). This plot was generated by dividing the data presented in (D) with the luminescence data in (B) to correct for changes in luciferase function with temperature and normalizing to the luminescence value at room temperature. When correcting for temperature, larger increases in luminescence over temperature is observed when split luciferase is fused to the 20 Å and 50 Å proteins compared to only the 20 Å or 50 Å proteins.

New Supplementary Figure 17E:

Supplementary Figure 17. Analysis of split luciferase reconstitution in response to domain dissolution via heating. (A) Raw luminescence data of split luciferase constructs in DOPC and ternary lipid membranes (42.5% 14:1 PC/27.5% DPPC/30% Chol). Luminescence values differ due to differences in expression. Values decrease with increasing temperature, as luciferase is less efficient at elevated temperatures as demonstrated by a luminescence vs temperature for soluble NanoBit (B). By normalizing by the initial value, we can compare how luminescence for each combination of proteins changes in (C) DOPC and (D) ternary lipid membranes (42.5% 14:1 PC/27.5% DPPC/30% Chol). For domain forming mixtures, transmembrane homodimers decrease more quickly than the heterodimer case. This suggests that homodimers interact less a result of lipid demixing and heterodimers are able to interact more and thus reconstitute luciferase. Data used to generate Fig. 3H. $n=3$, error bars represent the S. E. M. (E) An increase in normalized luminescence is also observed in for ternary lipid membranes (42.5% 14:1 PC/27.5% DPPC/30% Chol) containing the 20 and 50 Å protein relative to membranes containing either 20 Å-20 Å or 50 Å-50 Å proteins. This plot was generated by dividing the data presented in (D) with the luminescence data in (B) to correct for changes in luciferase function with temperature and normalizing to the luminescence value at room temperature. $n=3$, error bars represent the S. E. M.

Minor

comments:

1. At the first mention of “thin, medium, and thick” membranes in the Results, it would be helpful to add in the estimated thicknesses again to aid those readers that might skip over the introduction.

We have added the estimated thicknesses in the results, as well as a supplementary table.

2. The cartoon in 2G is the opposite of what is shown in 2H. Do proteins functionally insert into thinner membranes or thicker membranes? Maybe I'm confused considering the calculations for the y-axis of 2F and 2H are not given anywhere.

Proteins functionally insert into both thin and thick membranes, but you are correct in that we get more dye transfer into thick membranes in Figure 2H. We have switched the schematic around. Thanks for the suggestion.

Further, we have added equations describing how Fig. 2F and 2H were calculated to supplement descriptions within the methods to increase clarity.

3. The manufacturer of all materials needs to be listed (especially the antibodies used as they are highly variable).

We have checked the materials and added missing information, including the product numbers for primary antibodies used in this work.

4. To fully understand Fig S13, the exact nature of the lipids and FRET pair needs to be stated. NBD-labeled lipids are never mentioned in the Methods. Are these head group-labeled or tail-labeled? Is it PC or PE-rhodamine?

Thanks for pointing this out. NBD-labeled lipids were added to the methods and more explicit statements were added to the figure caption and supplementary table. NBD and Rhodamine were both conjugated to the headgroup of PE lipids (18:1 and 16:0).

5. Again, the lipid compositions of the modeled systems needs to be listed clearly in the Results and figure legends. This statement is unacceptable: "We simulated membrane interactions with thin (20 Å) and thick (50 Å) proteins using coarse-grained MD simulations of lipid composition comparable to the experimental system." What is comparable? How is the "mol Fraction DyPC in contact with protein" calculated in Fig 3A? Perhaps the authors can instead calculate number of DyPC or DPPC in contact with protein.

We have updated this Figure caption (as well as all others) to clearly state lipid compositions for both simulated and experimental membranes. We furthermore added a table in the methods section summarizing all simulated systems.

The mol Fraction DyPC in contact with protein is calculated by determining the time-averaged membrane composition in contact with each protein and reporting this as a fraction (DyPC lipids divided by total number of lipids, including DPPC, DyPC and Cholesterol). This is effectively the same as reporting the total number of DyPC molecules in contact with each protein and is noted in the Methods section of the manuscript.

Figure 3A caption now reads:

The 20 and 50 Å thick proteins were simulated in 42 mol% DyPC/28 mol% DPPC/30 mol% Cholesterol membranes. mol Fraction of DyPC in contact with the protein was determined by considering the time-averaged membrane composition of lipids in the first shell around each protein. These simulations indicate the 20 Å hairpin proteins interact

more with the shorter lipid, DyPC, compared to the 50 Å protein. As temperature increases, the protein-DyPC contacts shift towards the average membrane composition. The dotted line indicates the actual composition of DyPC, 42 mol%.

New Table 1 summarizing simulation results:

Table 1. Summary of systems simulated.

Measurement	Figure	Protein	Membrane composition	Simulation time [μs] x Replicates	Total simulation time per lipid composition [μs]
Membrane thickness and compression	Figure 1B, C, F; Supplementary Figure 1	Single protein (20, 28, 50 Å proteins)	DOPC, DYPC, or DGPC	6 μs x 3	18 μs
Lipid-protein interactions	Figure 3A	Single protein (20 or 50 Å proteins)	42 DYPC:28 DPPC: 30 CHOL	6 μs x 3	18 μs
Protein-protein interactions	Supplementary Figure 15	20 Å – 20 Å proteins	DOPC or 42 DYPC:28 DPPC: 30 CHOL	6 μs x 3	18 μs
Protein-protein interactions	Supplementary Figure 15	50 Å – 50 Å proteins	DOPC or 42 DYPC:28 DPPC: 30 CHOL	6 μs x 3	18 μs
Protein-protein interactions	Figure 3F, Supplementary Figure 12, Supplementary Figure 15	20 Å – 50 Å proteins	DOPC or 42 DYPC:28 DPPC: 30 CHOL	6 μs x 9	54 μs

6. Why don't the values for the domains 47C in Fig S15 add up to 1 like they should? Something seems to be off in this plot. Perhaps a finer binning than 5nm would help to make the point more clear. What do these distances represent in terms of the size of the box modeled. Is 5nm "close"? If so, how many lipids away?

This plot reports protein center of mass distances. That means the distance can never go to zero as the center of mass lies inside the protein. Below is a snapshot with scale bars in a DOPC membrane (gray) with 20 Å protein (cyan) and 50 Å protein (pink). Scale bar is 1 nm. About 3 nm center to center distance seems to about the minimum we can reach without changes to the fold (which we don't allow). Thus, also a distance of about 4.5 nm (shown in the right) allows for direct protein-protein interactions without any lipid in-between the residues. We have replotted this data with a bin size of 3 nm to enhance resolution.

Replotted Supplementary Figure 15B:

Supplementary Figure 15. MD simulations predict protein-protein in synthetic membranes. (A) Simulations of homotypic and heterotypic pairs of the 50 Å and 20 Å proteins were run in single component (DOPC, no domains) or tertiary membranes (42.5 mol% DyPC/27.5 mol% DPPC/30 mol% Cholesterol, domains). The changes in protein association were quantified by calculating the pseudo free energy difference between the protein-protein tertiary and single component membranes as $-\log(N_{\text{bound, ternary}}/N_{\text{bound, single}})$, where N is the number of states which we considered bound. Proteins were considered bound if their center-to-center distance was below 3 nm. Standard errors were calculated from simulation replicates. Negative values represent enhanced protein-protein contact formation in ternary membranes compared to single component membranes. These data demonstrate that proteins of different transmembrane domain lengths are farther apart in the ternary lipid mixture relative to the homogenous membrane but proteins of equal transmembrane domain length are closer together in ternary membranes relative to homogenous membranes.¹ (B) Protein-protein distance of 20 and 50 Å proteins in DOPC and domain forming (42.5 mol% DyPC/27.5 mol% DPPC/30 mol% Cholesterol) lipid mixtures at 25°C and 47°C. At 25 °C, protein-protein distance between the 20 and 50 Å hairpin is on average smaller in homogenous, single component DOPC membranes compared to membranes composed of 42.5 mol% DyPC/27.5 mol% DPPC/30 mol% Cholesterol. At 47°C, protein-protein distance decreases in membranes composed of 42.5 mol% DyPC/27.5 mol% DPPC/30 mol% Cholesterol due to increased lipid mixing. Histograms were generated from 3 independent simulations. Bin size is 3 nm, the approximate protein center to center.

7. What is the concentration of BSA used in coating the glass coverslips for the GUVs?

1% BSA in PBS was used to block coverslips for GUV visualization. This has been added to the methods.

Reviewer #3 (Remarks to the Author):

The manuscript by Peruzzi et al demonstrates how hydrophobic mismatch can be exploited to tune protein integration and organization in model membranes using de novo protein design, molecular dynamics simulations, and cell-free systems. The hydrophobic mismatch is among the effective membrane-mediated interactions that are also believed to be involved in cellular membrane organization in living cells. Membrane lateral organization and mechanisms that govern their organization are interesting and important topics. Combining these techniques is a robust scheme to explore such mechanisms. However, prior to making any decision regarding the acceptance of the article, I believe there are several important points that should be addressed. Therefore, I will reconsider my decision once I have received answers to the following comments.

My main problem is the presentation of the simulation results.

Molecular dynamics simulations have been performed by the authors, but there is no snapshot of the simulated systems (except for Figure SI-12, which is also unclear). As an example, the cartoon figures in figures 1 and 3 (in particular figure 1A) should be replaced by snapshots of the simulated system. A table should be provided (either in the SI or in the main text) indicating all simulated systems and the amount of time that has been simulated.

We thank the reviewer for these suggestions. We have added additional snapshots of simulations performed to calculate membrane deformations in Figure 1 and protein-lipid and protein-protein interactions in Figure 3. We have also added a new image for Supplementary Figure 12 to enhance clarity. We have not added these snapshots to the main figures, however, as we would like to still emphasize how the proteins were designed in Figure 1A and believe cartoons in Fig. 1B, 3A, and 3E enhance the understanding of what was characterized and found in a more simplistic way, making it more immediately understandable to a wider audience. Further, we have added a table to the methods summarizing all the simulated systems and amount of time that was simulated.

Below please find the updated Figures and Table.

Supplementary Figure 1. Membranes must deform to accommodate hydrophobic mismatch. Using MD simulations, the thickness of the membrane was recorded as a function of distance from the protein. (A) Snap shot of simulation of 20 Å thick protein in DOPC membranes shows local membrane deformation. This analysis was performed for the 20 (B), 28 (C), and 50 Å (D) proteins in thin (DYPC), medium (DOPC), and thick (DGPC) membranes. Horizontal dotted lines represent the hydrophobic thickness of the protein. Data presented in Fig. 1C was generated by subtracting the membrane thickness at 70 Å from membrane thickness at 10 Å from the protein center.

Supplementary Figure 12. Simulation of 20 and 50 Å proteins in phase separating membranes. DYPC lipid (red) nucleates around the 20 Å hairpin protein (pink). The 50 Å protein (blue) is in contact with DPPC (grey) and cholesterol (gold) more often than DYPC (red). Further, proteins are apart from one another. Representative image is from Movie S1. The membrane is composed of 42.5 mol% DYPC/27.5 mol% DPPC/30 mol% Cholesterol.

Table 1. Summary of systems simulated.

Measurement	Figure	Protein	Membrane composition	Simulation time [μs] x Replicates	Total simulation time per lipid composition [μs]
Membrane thickness and compression	Figure 1B, C, F; Supplementary Figure 1	Single protein (20, 28, 50 Å proteins)	DOPC, DYPC, or DGPC	6 μs x 3	18 μs
Lipid-protein interactions	Figure 3A	Single protein (20 or 50 Å proteins)	DYPC:DPPC:CHOL	6 μs x 3	18 μs
Protein-protein interactions	Supplementary Figure 15	20 Å – 20 Å proteins	DOPC or DYPC:DPPC:CHOL	6 μs x 3	18 μs
Protein-protein interactions	Supplementary Figure 15	50 Å – 50 Å proteins	DOPC or DYPC:DPPC:CHOL	6 μs x 3	18 μs
Protein-protein interactions	Figure 3F, Supplementary Figure 12, Supplementary Figure 15	20 Å – 50 Å proteins	DOPC or DYPC:DPPC:CHOL	6 μs x 9	54 μs

As it is now, it seems that the simulations are only 6 microseconds long. Does 6 microseconds is enough to observe phase separation in ternary mixture systems? Because Figure SI-12 does not show a phase-separated membrane. Moreover, in Figure SI-12 both DPPC and CHOL are shown with the same color which makes it even more unclear.

We would like to emphasize that the simulation membranes are not phase separated in the absence of protein. We assemble membranes and then insert models of the protein constructs in the membrane. Our simulations consider low protein converge, consistent with experiment. As the membranes are fluid, any perturbations induced by the proteins should decay away from the protein with a typical length scale set by the membrane thickness. Thus, even with protein inclusions the lipid membranes should not be phase separated sufficiently far away from the membrane protein. Instead, there is a small domain of lipids around the protein that is distinguishable and different from the average membrane composition. We have quantified this effect in the manuscript (Fig 3A) by considering the membrane composition of lipids in the first shell around the protein. Depending on the protein hydrophobic mismatch different lipid domain compositions were found. As the domains are small compared to the simulation box, we found the microsecond long simulations sufficient for convergence. This situation is also shown more clearly in the new snapshot in Fig S12 where we now have included the cholesterol molecules with separate colors. Additionally, we have performed additional replicates and new simulation with a total time of more than 50 μ s (Table 1).

Supplementary Figure 12. Simulation of 20 and 50 Å proteins in phase separating membranes. DYPG lipid (red) nucleates around the 20 Å hairpin protein (pink). The 50 Å protein (blue) is in contact with DPPC (grey) and cholesterol (gold) more often than DYPG (red). Further, proteins are apart from one another. Representative image is from Movie S1. The membrane is composed of 42.5 mol% DYPG/27.5 mol% DPPC/30 mol% Cholesterol.

Fig 4A has been mentioned in the below sentence but I could not find it. "For simulations shown in Fig. 4A, membranes were composed of 138 DPPC, 92 DYPG and 99 cholesterol molecules per leaflet"

Thank you for pointing this out. Figure 4A was mistakenly referenced. This has been fixed to refer to Figure 3A.

In this study, the authors investigated the effects of temperature change using the Martini model. The authors should provide some articles that demonstrate the Martini model is temperature transferable or, if not, explain why this won't be relevant to their study.

The reviewer is right that the MARTINI model is a coarse-grain model which does not necessarily reflect each physical parameter realistically. We considered the dissolution of protein induced lipid domains by increases in temperature. MARTINI has been previously used to simulate lipid only membranes at varying temperatures and was found to capture temperature induced phase changes, at least qualitatively^{6,7}. We found similar behavior between our experiments at varying temperatures and the behavior of the MARTINI model (Fig. 3C), further indication that temperature effects are well represented in the model. It is true however, that the domains around the 50 Å protein appeared in simulation more stable than in experiments with temperature increases. We have added a discussion of this discrepancy to the main text.

“The FRET data obtained as a function of temperature for the 20 Å protein mirrors our simulation data in Fig. 3A. MARTINI simulations have been previously used to simulate lipid only membranes at varying temperatures and were found to capture temperature induced phase changes, at least qualitatively.^{6,7} However, domains around the 50 Å protein in the simulation appear to be more stable than in the synthetic membrane experiments, likely due to different parameters between the MARTINI model and experimental conditions.

The authors write “However, it has also been hypothesized that more passive lipid-protein interactions can drive inter- and intramembrane protein organization” and cites several articles that only a few are relevant and relevant existing literature are missing. For example, see several works by Johannes group: Johannes et al, Clustering on membranes: fluctuations and more; Trends in cell biology 28 405-415 2018 Pezeshkian et al, Mechanism of Shiga toxin clustering on membranes; ACS nano 11 314-324 (2017) Arumugam et al, Ceramide structure dictates glycosphingolipid nanodomain assembly and function, Nature communications 12, 1-12 (2021)

We thank the reviewer for bringing this literature to our attention. We have cited the following papers:

7. Johannes, L., Pezeshkian, W., Ipsen, J. H. & Shillcock, J. C. Clustering on Membranes: Fluctuations and More. *Trends Cell Biol* **28**, 405–415 (2018).
8. Arumugam, S. *et al.* Ceramide structure dictates glycosphingolipid nanodomain assembly and function. *Nature Communications* 2021 12:1 **12**, 1–12 (2021).
9. Nowakowski, P., Stumpf, B. H., Smith, A.-S. & Maciołek, A. Demixing of homogeneous binary lipid membranes induced by protein inclusions. *Phys Rev E* **107**, 54120 (2023).
10. MacHta, B. B., Veatch, S. L. & Sethna, J. P. Critical casimir forces in cellular membranes. *Phys Rev Lett* **109**, 138101 (2012).
11. Soubias, O., Teague, W. E., Hines, K. G. & Gawrisch, K. Rhodopsin/Lipid Hydrophobic Matching—Rhodopsin Oligomerization and Function. *Biophys J* **108**, 1125–1132 (2015).
37. Shelby, S. A., Castello-Serrano, I., Wisser, K. C., Levental, I. & Veatch, S. L. Membrane phase separation drives responsive assembly of receptor signaling domains. *Nature Chemical Biology* 2023 19:6 **19**, 750–758 (2023).
43. Kim, T. *et al.* Influence of hydrophobic mismatch on structures and dynamics of gramicidin A and lipid bilayers. *Biophys J* **102**, 1551–1560 (2012).

45. Alves, I. D., Salamon, Z., Hruby, V. J. & Tollin, G. Ligand modulation of lateral segregation of a G-protein-coupled receptor into lipid microdomains in sphingomyelin/phosphatidylcholine solid-supported bilayers. *Biochemistry* **44**, 9168–9178 (2005).

Through the addition of the new citations (listed above) and additional text, we have tried to better acknowledge previous work and more clearly articulate that novelty of this study lies in the cell-free folding and expression data, the one-pot assembly of membranes with distinct protein incorporation and corresponding function based on hydrophobic mismatch alone, and the control over the protein-protein interactions in a single synthetic membrane through the use and dissolution of membrane phase separated domains.

Revision References:

1. Katira, S., Mandadapu, K. K., Vaikuntanathan, S., Smit, B. & Chandler, D. Pre-transition effects mediate forces of assembly between transmembrane proteins. *Elife* **5**, (2016).
2. Heberle, F. A. *et al.* Direct label-free imaging of nanodomains in biomimetic and biological membranes by cryogenic electron microscopy. *Proc Natl Acad Sci U S A* **117**, 19943–19952 (2020).
3. Nagle, J. F. & Tristram-Nagle, S. Structure of lipid bilayers. *Biochim Biophys Acta* **1469**, 159 (2000).
4. Steinküher, J. *et al.* Improving cell-free expression of membrane proteins by tuning ribosome co-translational membrane association and nascent chain aggregation. *bioRxiv* 2023.02.10.527944 (2023) doi:10.1101/2023.02.10.527944.
5. Choi, H. K. *et al.* Watching helical membrane proteins fold reveals a common N-to-C-terminal folding pathway. *Science (1979)* **366**, 1150–1156 (2019).
6. Liu, Y., Barnoud, J. & Marrink, S. J. Gangliosides Destabilize Lipid Phase Separation in Multicomponent Membranes. *Biophys J* **117**, 1215–1223 (2019).
7. Carpenter, T. S. *et al.* Capturing Phase Behavior of Ternary Lipid Mixtures with a Refined Martini Coarse-Grained Force Field. *J Chem Theory Comput* **14**, 6050–6062 (2018).

Reviewers' Comments:

Reviewer #1:

Remarks to the Author:

I do appreciate the authors' efforts for testing their simulations about phase separation induced by protein insertion. I agree that in the current stage of their experiments, these new results are not significant enough to be added in the manuscript.

The new simulations (Fig. Supp. 15) are also fine.

With the new additions, I think that the paper is now suitable for publication, and should be of interest for biophysicists and cell biologists.

Reviewer #2:

Remarks to the Author:

The authors have sufficiently addressed all of my concerns. This is a beautiful, well-executed manuscript.

Reviewer #3:

Remarks to the Author:

The authors have not adequately addressed all my concerns. Below, I will outline what are still lacking.

1) It is fine, if the authors want to keep Figure 1 in its current form. However, simulation snapshots, much like the cartoon image should be incorporated for example in the SI. The added snapshot to Figure SI-1 does not show much about the main focus of the manuscript, i.e., changes in the membrane thickness in proximity of the protein. What I would like to see is something like those depicted in Figure 1A, albeit derived from simulations (snapshot correspondence of Figure 1B or Figures SI-1B, C, D). I expect that the cartoon image might exaggerate the membrane thinning phenomena (which is fine to me) but my concern is that the current snapshot does not show any increase. Figure SI-1-B and C red lines indicate an increase, around 1-1.5 nm, in the membrane thickness within a 2.5 nm distance from the protein perimeter. But this does not appear in the snapshot. What simulated system the snapshot belongs to? I suggest that the authors provide snapshots of the final frame of all simulations in the SI. This should be straightforward and important for clarity and solidity of the results. As a suggestion to provide good visualizations of the effect, the authors could highlight the PO4 beads with a different color (the beads that were used for the thickness calculations). The rest of the lipids can be made transparent.

2) The added table is nice but still lacks the exact composition i.e., what is the concentration (or number) of each lipid in the system?

3) Finally, in response to one of my previous comments, the authors state that the membranes are not phase-separated in the absence of the proteins. Ternary mixtures of DPPC, CHOL, and DOPC show phase separation (lo/l_d phases) for a wide range of compositions and temperatures (that is the reason the composition is important to be reported in the table). What is the exact composition here? Note, that Martini model is rather a good model for capturing phase separations (see Figure 3 of Chem. Rev.2019, 119, 6184–6226).

The rest of the concerns have been addressed by the authors.

REVIEWER COMMENTS

Reviewer #3 (Remarks to the Author):

The authors have not adequately addressed all my concerns. Below, I will outline what are still lacking.

1) It is fine, if the authors want to keep Figure 1 in its current form. However, simulation snapshots, much like the cartoon image should be incorporated for example in the SI.

The added snapshot to Figure SI-1 does not show much about the main focus of the manuscript, i.e., changes in the membrane thickness in proximity of the protein. What I would like to see is something like those depicted in Figure 1A, albeit derived from simulations (snapshot correspondence of Figure 1B or Figures SI-1B, C, D). I expect that the cartoon image might exaggerate the membrane thinning phenomena (which is fine to me) but my concern is that the current snapshot does not show any increase. Figure SI-1-B and C red lines indicate an increase, around 1-1.5 nm, in the membrane thickness within a 2.5 nm distance from the protein perimeter. But this does not appear in the snapshot. What simulated system the snapshot belongs to? I suggest that the authors provide snapshots of the final frame of all simulations in the SI. This should be straightforward and important for clarity and solidity of the results. As a suggestion to provide good visualizations of the effect, the authors could highlight the PO₄ beads with a different color (the beads that were used for the thickness calculations). The rest of the lipids can be made transparent.

We thank the reviewer for this suggestion. We agree that it would be helpful to include snapshots that more closely mirror the cartoon (which is exaggerated to highlight the average change in membrane thickness we measured), however, this is difficult due to the (1) dynamic nature of membranes on short time scales and (2) density of lipids immediately adjacent to the protein. We have adjusted the image in Supplementary Figure 1 (below) to better reflect membrane deformations around the proteins. Specifically, Supplementary Fig. 1a now shows a snapshot of the 20 Å protein in the thickest membrane system, DGPC, with only the headgroup beads shown. From this image, the deformation of the lipids can be better seen. We have only included this image, however, as the bending of membranes is generally hard to visualize on short timescales (as is captured in a snapshot) because the location of individual lipids is dominated by their "protrusion" motion; that is, they move up and down and do not form a smooth surface. This phenomenon has been previously noted¹. Therefore, the average distances between the head group beads are reported in Fig. 1 and Supplementary Fig. 1.

Further, we would like to note that the largest deformations are observed just at the edge of the protein. Due to these larger deformations, lipids appear to be less dense adjacent to the protein which also makes it difficult to observe the deformations in single snapshots.

Supplementary Figure 1. Membranes must deform to accommodate hydrophobic mismatch. Using MD simulations, the thickness of the membrane was recorded as a function of distance from the protein. **(A)** Snapshot of the simulation of the 20 Å thick protein in DGPC membrane shows local membrane deformation. This corresponds to the thick membrane (red line) in **(B)**. The lipid headgroups are shown in red. This analysis was performed for the 20 **(B)**, 28 **(C)**, and 50 Å **(D)** proteins in thin (DYPC), medium (DOPC), and thick (DGPC) membranes. Horizontal dotted lines represent the hydrophobic thickness of the protein. Data presented in Fig. 1C was generated by subtracting the membrane thickness at 70 Å from membrane thickness at 10 Å from the protein center.

Attached is a single representative snapshot of DGPC with 422 (the thinnest construct). Only the headgroup beads of the model are shown. We believe one can see the deformation. There is also some "noise" in the headgroup region. That's why it's hard to visualize the shape changes in a single snapshot for the other combinations because we show that the thickness changes are often below a nanometer. That is on the order of the protrusion motion; That's why we average, and the distance between the red beads is what is calculated on average for the main figure. We should also note that the largest deformations are observed just at the edge of the protein. These states are not very populated by lipids because they have to deform so much (another way to say this is that the membrane density is low right at the interface of the protein). That's why it's additionally unlikely to see the largest deformation in these single snapshots.

2) The added table is nice but still lacks the exact composition i.e., what is the concentration (or number) of each lipid in the system?

We thank the reviewer for this suggestion. We have added the number of lipids per leaflet for all simulations in the table, in addition to the materials and methods section.

New Table:

Supplementary Table 3. Table of lipid compositions used.

Figure	Panel	Composition
Fig. 1	b	DYPC (di-C12:1-C14:1 PC), DOPC (18:1 PC), DGPC (di-C20:1-C22:1) (simulation); 216 MARTINI lipids per leaflet
	c	DYPC, DOPC, DGPC (simulation); 216 MARTINI lipids per leaflet
	e	Water, 14:1 PC, DOPC, 22:1 PC
	f	Comparison of simulated membrane compression in DYPC, DOPC, DGPC to GFP fluorescence in 14:1 PC, DOPC, 22:1 PC membranes
	g	Water, 14:1 PC, DOPC, 22:1 PC
Fig. 2	c	14:1 PC
	d	22:1 PC
	f	14:1 PC and 22:1 PC
	h	14:1 PC and 22:1 PC
Fig. 3	a	42 mol% DYPC/28 mol% DPPC (16:0 PC)/30 mol% Cholesterol (simulation) or 138 DYPC, 92 DPPC and 99 cholesterol molecules per leaflet
	c	42.5 mol% 14:1 PC/27.5 mol% DPPC/30 mol% Cholesterol
	d	42.5 mol% 14:1 PC/27.5 mol% DPPC/30 mol% Cholesterol
	f	42 mol% DYPC/28 mol% DPPC/30 mol% Cholesterol and DOPC (simulation) or 138 DYPC, 92 DPPC and 99 cholesterol molecules per leaflet and 326 DOPC molecules per leaflet
	h	42.5 mol% 14:1 PC/27.5 mol% DPPC/30 mol% Cholesterol and DOPC
Supplementary Fig. 1	b-d	DYPC, DOPC, DGPC (simulation); 216 MARTINI lipids per leaflet
Supplementary Fig. 2	a-h	Water, 14:1 PC, DOPC, 22:1 PC
Supplementary Fig. 3		99.9 mol% DOPC, 0.1 mol% 18:1 PE Cy5.5
Supplementary Fig. 4		Water, 14:1 PC, DOPC, 22:1 PC
Supplementary Fig. 5		Water, 14:1 PC, DOPC, 22:1 PC
Supplementary Fig. 6		Water, 14:1 PC, DOPC, 22:1 PC
Supplementary Fig. 7		Water, 14:1 PC, DOPC, 22:1 PC
Supplementary Fig. 8		DOPC

Supplementary Figure 9		99.9 mol% 14:1 PC, 0.1 mol% 18:1 PE Cy5.5; 99.9 mol% 22:1 PC, 0.1 mol% 18:1 PE Rhodamine
Supplementary Figure 10	a, b	99.9 mol% 14:1 PC, 0.1 mol% 18:1 PE Cy5.5; 99.9 mol% 22:1 PC, 0.1 mol% 18:1 PE Rhodamine
Supplementary Figure 11	a	42.5 mol% 14:1 PC/27.5 mol% DPPC/30 mol% Chol + 0.1 mol% 18:1 PE Rhodamine
	b	40 mol% 14:1 PC/40 mol% DPPC/20 mol% Chol + 0.1 mol% 18:1 PE Rhodamine
Supplementary Figure 12		42 mol% DYPC/28 mol% DPPC/30 mol% Cholesterol (simulation) or 138 DYPC, 92 DPPC and 99 cholesterol molecules per leaflet
Supplementary Figure 13	b	DOPC and 42.5 mol% 14:1 PC/27.5 mol% DPPC/30 mol% Cholesterol with 0.1 mol% 18:1 PE Rhodamine and either 0.1 mol% 18:1 PE or 16:0 PE NBD
Supplementary Figure 14	a	99.9 mol% DOPC, 0.1 mol % 18:1 PE Rhodamine
	b-c	42.5 mol% 14:1 PC/27.5 mol% DPPC/30 mol% Cholesterol with 0.1 mol% 18:1 PE Rhodamine
Supplementary Figure 15		42 mol% DYPC/28 mol% DPPC/30 mol% Cholesterol and DOPC (simulation) or 138 DYPC, 92 DPPC and 99 cholesterol molecules per leaflet and 326 DOPC molecules per leaflet
Supplementary Figure 16	b	DOPC and 42.5 mol% 14:1 PC/27.5 mol% DPPC/30 mol% Cholesterol (Domains)
Supplementary Figure 17	a-e	DOPC and 42.5 mol% 14:1 PC/27.5 mol% DPPC/30 mol% Cholesterol (Domains)

Materials and Methods Section:

“Data in panel Fig 1B, C was obtained for single component membranes with 216 MARTINI DyPC, DOPC or DGPC lipids per leaflet.”

“For simulations shown in Fig. 3A, membranes were composed of 138 DYPC, 92 DPPC and 99 cholesterol molecules per leaflet. In a radial selection around the protein center of mass, corresponding to the first layer of surrounding lipid molecules, individual lipid types were determined. The time average of detected lipids, then determined average membrane composition around the protein center at varying temperatures. For Fig. 3F the same DPPC:DyPC:cholesterol membranes as above were compared to membranes with 326 DOPC lipids per leaflet. Both membranes contained two copies of two different protein constructs. The distributions of center of mass protein-protein distances were determined for the two membrane compositions.”

3) Finally, in response to one of my previous comments, the authors state that the membranes are not phase-separated in the absence of the proteins. Ternary mixtures of DPPC, CHOL, and DOPC show phase separation (lo/l_d phases) for a wide range of compositions and temperatures (that is the reason the composition is important to be reported in the table). What is the exact composition here? Note, that Martini model is rather a good model for capturing phase separations (see Figure 3 of Chem. Rev.2019, 119, 6184–6226).

We thank the reviewer for this question. Simulated ternary lipid membranes were composed of 42 mol% DYPC/28 mol% DPPC/30 mol% Cholesterol or 138 DYPC, 92 DPPC and 99 cholesterol molecules per leaflet. By placing the composition of the ternary phase diagram from Figure 3 of Chem. Rev.2019, 119, 6184–6226, it can be seen that the composition is just on the boundary where two phases exist.

Further, plotting our composition on a ternary phase diagram generated from experiments by Veatch and Keller (Figure 4; Biophys J., 2003 Nov; 85(5): 3074–3083)², it can be seen that this composition also lies within the regime which is miscible at 25°C.

Both ternary phase diagrams support that our simulated and experimental membranes lie just above the miscibility point and should not form domains. Furthermore, these ternary diagrams were generated using DOPC, while we have used 14:1 PC experimentally. 14:1 PC possess shorter fatty acid chains than

DOPC and thus should have a lower phase transition temperature. In a ternary mixture, this composition that includes 14: 1 PC should further decrease the miscibility temperature compared to the DOPC/DPPC/Chol ternary membranes shown in the diagrams.

The rest of the concerns have been addressed by the authors.

References

1. Lipowsky, R. Remodeling of membrane compartments: Some consequences of membrane fluidity. *Biol Chem* **395**, 253–274 (2014).
2. Veatch, S. L. & Keller, S. L. Separation of Liquid Phases in Giant Vesicles of Ternary Mixtures of Phospholipids and Cholesterol. *Biophys J* **85**, 3074 (2003).

Reviewers' Comments:

Reviewer #3:

Remarks to the Author:

I recommend the current manuscript for publication.

Point by Point response (our response in blue)

Reviewer #3 (Remarks to the Author):

I recommend the current manuscript for publication.

We thank the reviewer for their thoughtful comments which improved the manuscript. In addition, we have updated the manuscript in order to comply with the md simulation checklist.